# Amplitude Uncertainties Everywhere All at Once

Henning Bahl[1], Nina Elmer[1], Tilman Plehn[1,2], and Ramon Winterhalder[3]

**1** Institut für Theoretische Physik, Universität Heidelberg, Germany
**2** Interdisciplinary Center for Scientific Computing (IWR), Universität Heidelberg, Germany
**3** TIFLab, Università degli Studi di Milano & INFN Sezione di Milano, Italy

January 7, 2026

## Abstract

Ultra-fast, precise, and controlled amplitude surrogates are essential for future LHC event generation. First, we investigate the noise reduction and biases of network ensembles and outline a new method to learn well-calibrated systematic uncertainties for them. We also establish evidential regression as a sampling-free method for uncertainty quantification. In a second part, we tackle localized disturbances for amplitude regression and demonstrate that learned uncertainties from Bayesian networks, ensembles, and evidential regression all identify numerical noise or gaps in the training data.

# 1  Introduction

Understanding the fundamental forces and particles that shape our universe demands increasingly precise theoretical predictions and experimental measurements. All across particle physics, even minor discrepancies between theory and data can hint at physics beyond the Standard Model. The LHC experiments are already producing vast amounts of extremely complex data, and this volume is expected to increase significantly with the upcoming High-Luminosity LHC (HL-LHC). Correspondingly, precise and reliable first-principle simulations are becoming a central challenge for particle theory.

Modern machine learning (ML) has emerged as a transformative tool for addressing this challenge [1, 2]. From accelerating phase-space sampling [3–12] to evaluating complex scattering amplitudes [13–25], generating full events [26–30], or simulating detectors with unprecedented speed [31–54], ML plays a crucial role in every aspect of a sophisticated simulation chain.

Many critical ML improvements include surrogate and generative models capable of learning and reproducing the complex structures found in collider data [55–57]. For high-precision tasks such as amplitude regression, it is essential that these models not only predict the mean with high accuracy but also provide a calibrated local uncertainty estimate. Even when uncertainties are not directly propagated into experimental analyses, they are vital to justify the replacement of traditional calculations with ML surrogates. In a complementary approach [58–60], potential biases or inaccuracies of the surrogate can be avoided at the price of a secondary reweighting.

A range of methods exists to learn uncertainties of ML predictions, including Bayesian neural networks (BNNs) [61–63], repulsive ensembles (REs) [64–66], and more recently, evidential deep learning [67–70]. Each method has its strengths and weaknesses in terms of computational efficiency, interpretability, and calibration quality. Previously [23], we demonstrated that neural surrogates can reproduce loop-induced scattering amplitudes with per-mille level precision, while also learning calibrated uncertainties. However, we found that repulsive ensembles, which are promising for capturing network uncertainty, did not adequately calibrate uncertainties in particular regions of phase space.

In this study, we aim to clarify whether repulsive ensembles can serve as reliable posterior estimators for amplitude regression. We examine how the repulsive kernel affects uncertainty calibration and propose improvements to mitigate biases as well as to obtain well-calibrated uncertainties. Beyond ensembles, we study evidential regression (ER) as an alternative method that avoids sampling over neural weights and instead places priors on the hyperparameters of the predictive likelihood. This offers an efficient way to disentangle systematic and statistical uncertainties, eliminating the need for large ensemble sizes. Moreover, we explore challenging scenarios involving threshold smearing and gaps in the training data — effects that mimic the numerical instabilities or incomplete coverage often encountered in amplitude calculations near physical thresholds. We compare different ways to learn uncertainties for clean and for smeared datasets, to assess how well these methods can capture and calibrate uncertainties under realistic conditions.

This paper is organized as follows: In Sec. 2, we summarize the methods for repulsive ensembles and evidential regression and introduce standard observables to quantify the calibration of the estimated uncertainty. In Secs. 3 and 4, we present detailed studies of repulsive ensembles and evidential regression, respectively. Results for various localized learning challenges, including smearing and gap scenarios, are presented in Secs. 5.

## 2  Probabilistic amplitude regression

We approach amplitude regression from a probabilistic perspective, allowing us to predict the amplitude $A(x)$ and its variance $\sigma^2(x)$. We describe the amplitude prediction for a given phase-space point $x$ as a distribution $p(A|x)$, which implicitly depends on the training data $D_{\text{train}} = \{A_{\text{train}}, x_{\text{train}}\}$. It reflects data and network uncertainties and is induced by a posterior $p(\theta|D_{\text{train}})$ over the neural network weights,

$$p(A|x) = \int \mathrm{d}\theta \, p(\theta|D_{\text{train}}) \, p(A|x,\theta) \approx \int \mathrm{d}\theta \, q(\theta) \, p(A|x,\theta) \,, \tag{1}$$

where $p(A|x,\theta)$ is the likelihood of observing amplitude $A$ at input $x$, given specific network parameters $\theta$. In the last step, we replace the true but usually intractable posterior with an approximate distribution $q(\theta)$, obtained either via variational inference or network ensembling. For notational simplicity, we omit the explicit conditioning on $D_{\text{train}}$ and simply write $p(A|x)$, $q(\theta)$, etc., with the dependence on training data understood implicitly.

Given $p(A|x)$, we compute both the mean amplitude prediction and the associated uncertainty as

$$A_{\text{NN}}(x) = \int \mathrm{d}A \, A \, p(A|x)$$

$$= \int \mathrm{d}\theta \, q(\theta) \overline{A}(x,\theta) \quad \text{with} \quad \overline{A}(x,\theta) = \int \mathrm{d}A \, A \, p(A|x,\theta) \,,$$

$$\sigma^2_{\text{tot}}(x) = \int \mathrm{d}A \, [A - A_{\text{NN}}]^2 \, p(A|x)$$

$$= \int \mathrm{d}\theta \, q(\theta) \Big[ \sigma^2(x,\theta) + \big(\overline{A}(x,\theta) - A_{\text{NN}}(x)\big)^2 \Big]$$

$$\equiv \sigma^2_{\text{syst}}(x) + \sigma^2_{\text{stat}}(x) \quad \text{with} \quad \sigma^2(x,\theta) = \int \mathrm{d}A \, \big[A - \overline{A}(x,\theta)\big]^2 \, p(A|x,\theta) \,. \tag{2}$$

In the last line, we split the total uncertainty into a systematic and a statistical part [2]. They are defined as

$$\sigma^2_{\text{syst}}(x) = \int \mathrm{d}\theta \, q(\theta) \, \sigma^2(x,\theta) \,,$$

$$\sigma^2_{\text{stat}}(x) = \int \mathrm{d}\theta \, q(\theta) \big[\overline{A}(x,\theta) - A_{\text{NN}}(x)\big]^2 \,. \tag{3}$$

What we denote as *systematic uncertainty* here is systematic in effect, but typically arises from stochasticity in the data. This component is irreducible and persists even with infinite data. However, the same systematic uncertainty may also absorb residual model mismatch, for example, due to limited network expressivity [23]. This contribution is, in principle, reducible and may decrease with improved network capacity or more suitable architectural choices. Different sources of systematic uncertainties cannot be separated from the structure of the learned systematics.

In contrast, what we denote as *statistical uncertainty* has a statistical origin. It may originate from either network-related causes, like too few training samples, or network-related limitations, like poor prior choices or underfitting. This component is reducible and vanishes in the limit of infinite data and optimal training. Importantly, the statistical uncertainty is also model-dependent, in the same way as the parameter-induced uncertainty in classical curve

fitting. For example, fitting a straight line to two data points yields a vanishing statistical uncertainty, whereas fitting a parabola results in an infinite uncertainty. This example illustrates that statistical uncertainty is not only data-driven but also strongly affected by the underlying model choice. More generally, our physics-inspired definition of uncertainties mixes data-related (aleatoric) and network-related (epistemic) components, and is mathematically defined by Eq.(3).

Our probabilistic model cannot capture systematic biases present within the data itself, such as a constant shift applied to all training examples. They remain undetectable without additional assumptions, external calibration, or domain knowledge. This form of uncertainty is often referred to as dataset bias or systematic data error in the literature.

To train the model and infer a predictive distribution that captures these uncertainty components, we need to specify a likelihood $p(A|x, \theta)$ and an optimization procedure for the parameters $\theta$. In the simplest case, we treat $\theta$ as fixed and minimize the negative log-likelihood over the training set

$$\mathcal{L} = -\Big\langle \log p(A|x, \theta) \Big\rangle_{x \sim D_{\text{train}}} . \tag{4}$$

The exact form of this training objective depends on the form of the likelihood $p(A|x, \theta)$.

## 2.1 Uncertainty estimation

To track a systematic uncertainty, our network has to predict not only a mean amplitude but also an input-dependent uncertainty that captures the intrinsic variability of the data. The simplest ansatz is a Gaussian likelihood with input-dependent mean and variance,

$$p(A|x, \theta) = \mathcal{N}(A|\overline{A}(x, \theta), \sigma^2(x, \theta)) , \tag{5}$$

Both, $\overline{A}(x, \theta)$ and $\sigma^2(x, \theta)$ are network outputs. This allows the network to extract systematic uncertainty directly from the data. The variance $\sigma^2(x, \theta)$ can, for example, reflect irreducible noise at each input point $x$ and corresponds to the systematic component $\sigma^2_{\text{syst}}(x)$ in Eq.(2). As part of Eq.(4) this likelihood defines the *heteroscedastic loss*

$$\mathcal{L}_{\text{het}} = \left\langle \frac{(A_{\text{train}}(x) - \overline{A}(x, \theta))^2}{2\sigma^2(x, \theta)} + \log \sigma(x, \theta) \right\rangle_{x \sim D_{\text{train}}} . \tag{6}$$

Although typical amplitude regression assumes noise-free labels, i.e. $A_{\text{train}}(x) = A_{\text{true}}(x)$, the heteroscedastic loss still allows us to capture the uncertainty, for instance, from limited network expressivity. Moreover, it can stabilize the training and lead to better accuracy and generalization compared to an MSE loss, as discussed below.

**Statistical uncertainties**

To fully capture the total uncertainty in Eq.(2), we must also account for the statistical uncertainty. It arises from our limited knowledge of the optimal network parameters due to finite training datasets or imperfect training. To model it, we return to $p(A|x)$ defined in Eq.(1). The integration over the network parameters uses an approximate form of $q(\theta)$. It allows us to estimate $\sigma_{\text{stat}}(x)$ by sampling over network configurations.

Several methods have been proposed to approximate the weight posterior $p(\theta|D_{\text{train}})$ via a tractable distribution $q(\theta)$. These include Bayesian neural networks (BNNs) [61–63, 65], which learn a posterior over weights using variational inference, and repulsive ensembles [64–

66], which approximate $q(\theta)$ through network replicas. Evidential regression [67,68] follows a different paradigm by predicting a distribution over possible outputs rather than sampling weights directly. It aims to capture both systematic and statistical uncertainties in a single forward pass.

In the present work, we concentrate on repulsive ensembles and evidential regression, which will be discussed in detail in Sections 3 and 4, respectively. For completeness, we also include BNNs as a benchmark in our studies of smeared data, but we do not revisit their methodology here, as BNNs have already been extensively studied in the context of amplitude regression in Refs. [18,23].

**Accuracy, calibration, and pulls**

We measure the accuracy of the network prediction using the local relative accuracy

$$\Delta(x) = \frac{A_{\mathrm{NN}}(x) - A_{\mathrm{true}}(x)}{A_{\mathrm{true}}(x)} \,. \tag{7}$$

To assess the calibration of the predicted uncertainty, we define the pull

$$t(x) = \frac{A_{\mathrm{NN}}(x) - A_{\mathrm{train}}(x)}{\sigma_{\mathrm{tot}}(x)} \,, \tag{8}$$

where $\sigma_{\mathrm{tot}}(x)$ captures the total predictive uncertainty at each phase-space point $x$. For a calibrated network the pull follows a unit Gaussian $\mathcal{N}(0,1)$ In the limit $\sigma_{\mathrm{stat}} \ll \sigma_{\mathrm{syst}}$ the pull simplifies to

$$t_{\mathrm{syst}}(x) = \frac{A_{\mathrm{NN}}(x) - A_{\mathrm{train}}(x)}{\sigma_{\mathrm{syst}}(x)} \,, \tag{9}$$

referred to as the systematic pull. To evaluate the calibration of the statistical uncertainty alone, we compare the network prediction to the noise-free truth

$$t_{\mathrm{stat}}(x) = \frac{A_{\mathrm{NN}}(x) - A_{\mathrm{true}}(x)}{\sigma_{\mathrm{stat}}(x)} \,. \tag{10}$$

Here, $A_{\mathrm{train}}(x)$ denotes the target values used during training, which may include numerical or stochastic noise (e.g. from Monte-Carlo integration), while $A_{\mathrm{true}}(x)$ refers to the underlying noise-free deterministic amplitude. In the absence of training noise, the two coincide, $A_{\mathrm{train}}(x) = A_{\mathrm{true}}(x)$. Hence, the systematic pull probes deviations with respect to the noisy training targets, whereas the statistical pull isolates fluctuations around the underlying truth. Assuming the network prediction has no systematic biases due to a lack in expressivity, this statistical pull of a calibrated network should also follow a unit Gaussian. Note that evaluating this quantity requires knowledge of $A_{\mathrm{true}}(x)$, which is only accessible for simulated data. A more detailed discussion of pull-based calibration can be found in Appendix D.3 of Ref. [65].

## 2.2 Extended likelihood parametrizations

So far, our discussion has been restricted to the standard heteroscedastic Gaussian likelihood of Eq.(6). While this formulation is well motivated from statistical principles, it might lead to unexpected behavior during training and is limited to unimodal distributions. To address these shortcomings, several modifications have been proposed in the literature. In the following, we discuss two such extensions. First, we consider the so-called natural parametrization of the Gaussian likelihood that has been suggested as a remedy for unstable optimization. Second, we outline how mixtures of Gaussians can be used to move beyond the single-Gaussian assumption and allow for multi-modal predictions.

**Natural parametrization**

It has been pointed out in the literature [71–74] that a heteroscedastic loss can behave unexpectedly during numerical optimization even though it is directly derived from statistical principles. This can be understood, if we remember how the heteroscedastic loss from Eq.(6) is parametrized in terms of the mean and variance

$$\mathcal{L}_{\text{het}} \equiv \mathcal{L}_{\text{het}}^{\bar{A},\sigma^2} = \left\langle \frac{(A_{\text{train}}(x) - \bar{A}(x,\theta))^2}{2\sigma^2(x,\theta)} + \log \sigma(x,\theta) \right\rangle_{x \sim D_{\text{train}}} . \quad (11)$$

Then, the gradients of the loss with respect to the mean and variance are

$$\nabla_{\bar{A}} \mathcal{L}_{\text{het}}^{\bar{A},\sigma^2} = \left\langle \frac{\bar{A}(x,\theta) - A_{\text{train}}(x)}{\sigma^2(x,\theta)} \right\rangle_{x \sim D_{\text{train}}}$$

$$\nabla_{\sigma^2} \mathcal{L}_{\text{het}}^{\bar{A},\sigma^2} = \left\langle \frac{\sigma^2(x,\theta) - (A_{\text{train}}(x) - \bar{A}(x,\theta))^2}{2\sigma^4(x,\theta)} \right\rangle_{x \sim D_{\text{train}}} , \quad (12)$$

where the scaling with $\sigma^{-2}$ in both gradients quickens learning for low-variance points and can thus be biased in regions where the mean predictions are poor. Here, a network may use high variance to explain poor mean estimates instead of improving them. This can create a 'rich-get-richer' dynamic, where points with lower predictive variance continuously provide the largest learning signal.

Among other solutions proposed in Refs. [72, 73], the most elegant solution is based on a simple reparametrization of the loss function. In Ref. [74], they propose to parametrize the heteroscedastic loss in terms of natural parameters

$$\eta_1(x,\theta) = \frac{\bar{A}(x,\theta)}{\sigma^2(x,\theta)} \quad \text{and} \quad \eta_2(x,\theta) = -\frac{1}{2\sigma^2(x,\theta)} \leq 0 , \quad (13)$$

which can be understood as the signal-to-variance ratio and the negative precision (inverse variance). With these parameters, the heteroscedastic loss can then be written as

$$\mathcal{L}_{\text{het}}^{\text{natural}} \equiv \mathcal{L}_{\text{het}}^{\eta_1,\eta_2} = \left\langle -\eta_2(x,\theta)\left(A_{\text{train}}(x) + \frac{\eta_1(x,\theta)}{2\eta_2(x,\theta)}\right)^2 - \frac{1}{2}\log(-2\eta_2(x,\theta)) \right\rangle_{x \sim D_{\text{train}}} . \quad (14)$$

Taking the gradients of this loss with respect to $\eta_i$ and then relating it to $\bar{A}$ and $\sigma^2$, we obtain

$$\nabla_{\eta_1} \mathcal{L}_{\text{het}}^{\eta_1,\eta_2} = \left\langle -\frac{\eta_1(x,\theta)}{2\eta_2(x,\theta)} - A_{\text{train}}(x) \right\rangle_{x \sim D_{\text{train}}}$$

$$= \left\langle \bar{A}(x,\theta) - A_{\text{train}}(x) \right\rangle_{x \sim D_{\text{train}}}$$

$$\nabla_{\eta_2} \mathcal{L}_{\text{het}}^{\eta_1,\eta_2} = \left\langle \frac{\eta_1(x,\theta)^2}{4\eta_2(x,\theta)^2} - \frac{1}{2\eta_2(x,\theta)} - A_{\text{train}}^2(x) \right\rangle_{x \sim D_{\text{train}}}$$

$$= \left\langle \sigma^2(x,\theta) - (A_{\text{train}}^2(x) - \bar{A}(x,\theta)^2) \right\rangle_{x \sim D_{\text{train}}} . \quad (15)$$

This reformulation is desirable because the gradients decouple the residuals for mean and variance. The mean is now updated by the prediction error, while the variance is updated by the mismatch between predicted and empirical second moments. In contrast to the standard parametrization, this avoids disproportionate weighting of low-variance points and leads to more balanced learning dynamics.

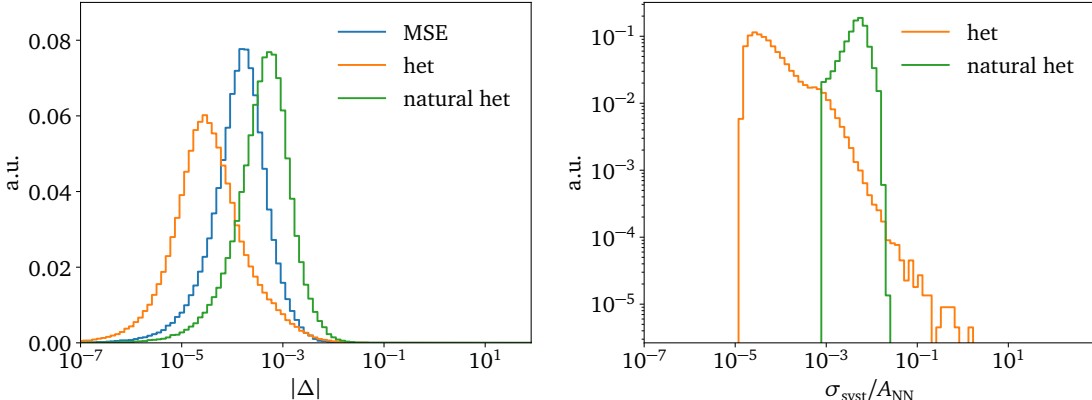

Figure 1: Comparison of MSE, heteroscedastic, and natural-heteroscedastic losses.

We tested whether the natural parametrization also improves optimization in practice. To this end, we trained our surrogates on the amplitude dataset introduced in Sec. 2.3 and compared the performance of three objectives: standard MSE, the conventional heteroscedastic loss, and the natural heteroscedastic loss, as shown in Fig. 1.

The most striking difference appears in the variance predictions (right panel). The natural parametrization leads to a much narrower distribution. This behavior is consistent with the fact that $\eta_2 = -1/(2\sigma^2)$ enforces a direct optimization of the precision, which tends to stabilize training by concentrating predictions around a typical variance value. In contrast, the conventional parametrization allows a broader spread of variance estimates.

Turning to accuracy (left panel), the conventional heteroscedastic loss performs best, followed by MSE, while the natural parametrization gives the worst results in the zero-noise case. This shows that heteroscedastic training is in principle helpful — since the conventional heteroscedastic loss outperforms MSE — but that the additional stabilization of the natural parametrization can come at the cost of reduced accuracy when no noise is present.

This outcome is not surprising. In the absence of noise, all residual variance originates from model uncertainty. In the natural parametrization, this model uncertainty is implicitly distributed between both natural parameters $(\eta_1, \eta_2)$, making optimization more convoluted than in the conventional parametrization, where mean and variance are disentangled more directly.

We further validated this behavior on a dataset with homogeneous noise (5% everywhere, not shown). In this case, both heteroscedastic variants perform similarly, while MSE remains inferior. These results suggest that the natural parametrization may only become advantageous in settings with heterogeneous noise across the training set, where its stricter control of variance could help balance the learning of mean and variance more effectively.

**Gaussian mixture model**

A limitation of the simple Gaussian likelihood in Eq.(5) is that it cannot capture multi-modality, heavy tails, or other non-Gaussian structures. A more expressive choice is therefore a Gaussian mixture model (GMM) with $K$ modes, where the likelihood is modeled as

$$p_{\text{GMM}}(A|x,\theta) = \sum_{k=1}^{K} \omega_k(x,\theta)\, \mathcal{N}\big(A\,\big|\,\overline{A}_k(x,\theta),\, \sigma_k^2(x,\theta)\big), \quad \text{with} \quad \sum_{k=1}^{K} \omega_k(x,\theta) = 1\,. \quad (16)$$

Here, the network has $3K$ outputs corresponding to the mixture means $\overline{A}_k(x,\theta)$, the variances $\sigma_k^2(x,\theta)$, and the mixture weights $\omega_k(x,\theta)$. The weights are typically parameterized through a softmax layer to guarantee $\omega_k \geq 0$ and proper normalization. The mean and variance of the GMM are then obtained from the mixture distribution as

$$\overline{A}_{\text{GMM}}(x,\theta) = \sum_{k=1}^{K} \omega_k(x,\theta)\,\overline{A}_k(x,\theta)\,,$$

$$\sigma_{\text{GMM}}^2(x,\theta) = \sum_{k=1}^{K} \omega_k(x,\theta)\left(\sigma_k^2(x,\theta) + \overline{A}_k^2(x,\theta)\right) - \left(\overline{A}_{\text{GMM}}(x,\theta)\right)^2\,. \tag{17}$$

In addition to the mixture mean and variance, one may also consider the maximum a posteriori (MAP) estimate, defined as

$$A_{\text{GMM}}^{\text{MAP}}(x,\theta) = \arg\max_A p_{\text{GMM}}(A|x,\theta)\,, \tag{18}$$

which corresponds to the largest mode of the predictive distribution. The MAP estimate can be more representative in cases where the mixture distribution is multi-modal and the mean lies in a region of low likelihood between distinct modes. In general, the MAP for a Gaussian mixture has no closed analytic form and must be obtained numerically, for example through grid evaluation or local optimization of $p_{\text{GMM}}(A|x,\theta)$. As part of Eq.(4), the Gaussian mixture model defines the negative log-likelihood loss

$$\mathcal{L}_{\text{GMM}} = -\left\langle \log\left[\sum_{k=1}^{K} \frac{\omega_k(x,\theta)}{\sqrt{2\pi\sigma_k^2(x,\theta)}} \exp\left(-\frac{(A_{\text{train}}(x) - \overline{A}_k(x,\theta))^2}{2\sigma_k^2(x,\theta)}\right)\right]\right\rangle_{x\sim D_{\text{train}}}\,. \tag{19}$$

In the special case $K = 1$, this expression reduces to the heteroscedastic loss in Eq.(6).

## 2.3  Dataset and network architecture

As in Ref. [23], we learn the loop-induced squared amplitude for the partonic process [15,18]

$$gg \to \gamma\gamma g\,, \tag{20}$$

as our benchmark. The dataset contains 1.1M unweighted events and is generated with SHERPA [75] and the NJET library [76]. The detector acceptance and object definition is mimicked by a set of basis cuts,

$$\begin{array}{lll} p_{\text{T},j} > 20 \text{ GeV} & |\eta_j| < 5 & R_{j\gamma,\gamma\gamma} > 0.4 \\ p_{\text{T},\gamma} > 40, 30 \text{ GeV} & |\eta_\gamma| < 2.37\,. \end{array} \tag{21}$$

If not mentioned otherwise, we use 70% of the dataset for training. 10% of the dataset are used for validation and selecting the best network; 20%, for testing. By default, we train for 1000 epochs. Figure 2 shows a histogram of the absolute magnitudes of the squared amplitudes in our dataset, illustrating the dynamic range of the regression task, spanning approximately five orders of magnitude.

In our previous study [23], we investigated various network architectures, including a simple multi-layer perceptron (MLP), a deep sets [77] inspired architecture, as well as a fully Lorentz and permutation-equivariant network architecture [20–22,78]. While increasing network complexity allows for more accurate amplitude predictions, it usually also increases the required training and evaluation time. In the same study, we have seen that a GELU activation

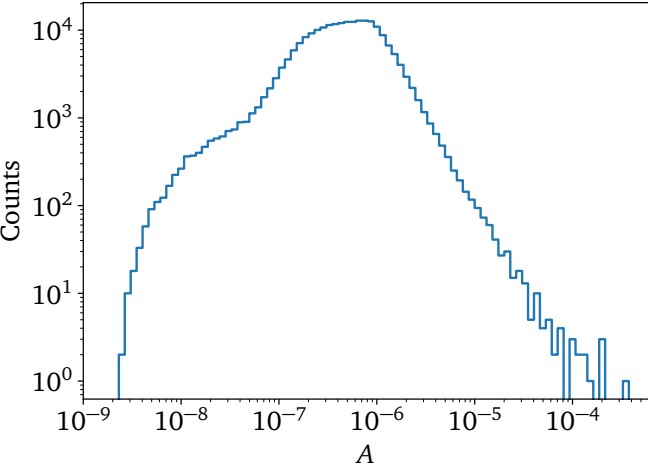

Figure 2: Distribution of the squared amplitudes $A$ in the dataset used for training and evaluation.

function gives the best results. Since the focus of this work is on the proper description of uncertainties, we stick to a simple MLP with carefully chosen input features and a GELU activation function. This is GPU efficient and provides sufficiently high accuracy. All hyperparameters are summarized in App. B.

As network input, we use the 4-vectors of the involved particles. We complement them with all possible (logarithmic) Mandelstam invariants

$$z_{ij} = \log p_i p_j \,. \tag{22}$$

We also apply a logarithmic transformation to the squared amplitudes to better capture the large variations across different phase-space points. All inputs and transformed targets are then standardized to mean zero and unit variance. The inverse transformation, including error propagation, is described in App. A.

# 3 (Repulsive) Ensembles

Repulsive ensembles approximate the posterior $p(\theta|D_{\text{train}})$ by training an ensemble of neural networks, where the distribution of ensemble members encodes the statistical uncertainty. To avoid a collapse of members to the same loss minimum, a repulsive term encourages parameter diversity around the loss minimum. For a training dataset of size $N$, evaluated in batches $B$, and $N_{\text{ens}}$ ensemble members, the loss is

$$\mathcal{L}_{\text{RE}} = \sum_{i=1}^{N_{\text{ens}}} \left[ -\frac{1}{B} \sum_{b=1}^{B} \log p(A|x_b, \theta_i) + \frac{\beta}{N} \frac{\sum_{j=1}^{N_{\text{ens}}} \mathcal{K}(\overline{A}(x, \theta_i), \hat{A}(x, \theta_j))}{\sum_{j=1}^{N_{\text{ens}}} \mathcal{K}(\hat{A}(x, \theta_i), \hat{A}(x, \theta_j))} + \frac{|\vec{\theta}_i|^2}{2N\sigma_{\text{prior}}^2} \right] . \quad (23)$$

Here, $\mathcal{K}$ is a kernel function, typically a radial basis function (RBF) that measures the similarity between predictions of ensemble members. The hat symbol denotes a stop-gradient operation, which prevents backpropagation through the comparison targets. The coefficient $\beta$ controls the strength of the repulsive interaction; we use $\beta = 1$ unless stated otherwise. The final term acts as a weight decay, corresponding to a Gaussian prior with standard deviation $\sigma_{\text{prior}}$ on the network weights; we use $\sigma_{\text{prior}} = 1$. For a detailed derivation of this loss and its connection to Bayesian inference, we refer to Refs. [2, 23].

## 3.1 Impact of the repulsive kernel

For a jointly trained ensemble, we would like to know what the influence of the repulsive prefactor $\beta$, defined in Eq.(23), on the training and uncertainty estimate is. Therefore, we vary the repulsive prefactor $\beta$ and simultaneously the number of training points $N_{\text{train}}$

$$\beta = \{10^{-5}, 0.01, 0, 1, 10\}, \qquad N_{\text{train}} = \{1.2\%, 3.4\%, 10.3\%, 30.5\%, 80\%\} \times 1.1 \cdot 10^6, \quad (24)$$

where the quoted values of $N_{\text{train}}$ arise from choosing uniformly spaced points in log space, while keeping the size of the test and validation data set fixed. Figure 3 shows the results for these different sets of $\beta$ and $N_{\text{train}}$. In the upper panel of plots, we observe a spread in the relative size of $\sigma_{\text{syst}}/A$ for smaller training data sets. This spread vanishes for $N_{\text{train}}$ larger than $10^5$ points, or 10% of the full dataset. For the relative statistical uncertainty $\sigma_{\text{stat}}/A$, this spread is slightly smaller for smaller training data sets, but vanishes again once we use more than 10% of the data for training purposes. The lower panel displays the mean accuracy $\langle \Delta \rangle$ of the ensemble prediction and the mean value for the systematic pull, $\langle t_{\text{syst}} \rangle$. For smaller training data sets, using less than 30% of the complete data set for training, we observe larger error bands for $\langle \Delta \rangle$. However, the results still agree with zero. These error bands are obtained by training the ensemble set up multiple times. These larger error bands for smaller $N_{\text{train}}$ can lead to a potential bias in the ensemble predictions towards non-zero values. It can also be observed that the choice of $\beta$ has only a small impact on the bias of $\langle \Delta \rangle$, since a spread is observed for every choice of $\beta$. Overall, when taking the results for every $N_{\text{train}}$ into account, the spread of the error bands is the most negligible for $\beta = 0$. The behavior of the ensemble towards a bias in its prediction will be discussed further in Sec. 3.2. The mean pull $\langle t_{\text{syst}} \rangle$ also fluctuates for smaller $N_{\text{train}}$ for all choices of $\beta$, but again stabilizes for larger $N_{\text{train}}$.

These observations suggest that the impact of the repulsive kernel is only visible for small training data sets. If the size of the training set surpasses 10% of the overall data set, the impact of $\beta$ becomes negligible. For the relative uncertainties $\sigma_{\text{syst}}/A$ and $\sigma_{\text{stat}}/A$, the effect vanishes completely, while for $\langle \Delta \rangle$ it is getting smaller and completely disappears when including more than 30% of the data for training. The spread observed for smaller amounts of data used for training purposes is related to the training dynamics of the system rather than the spread of

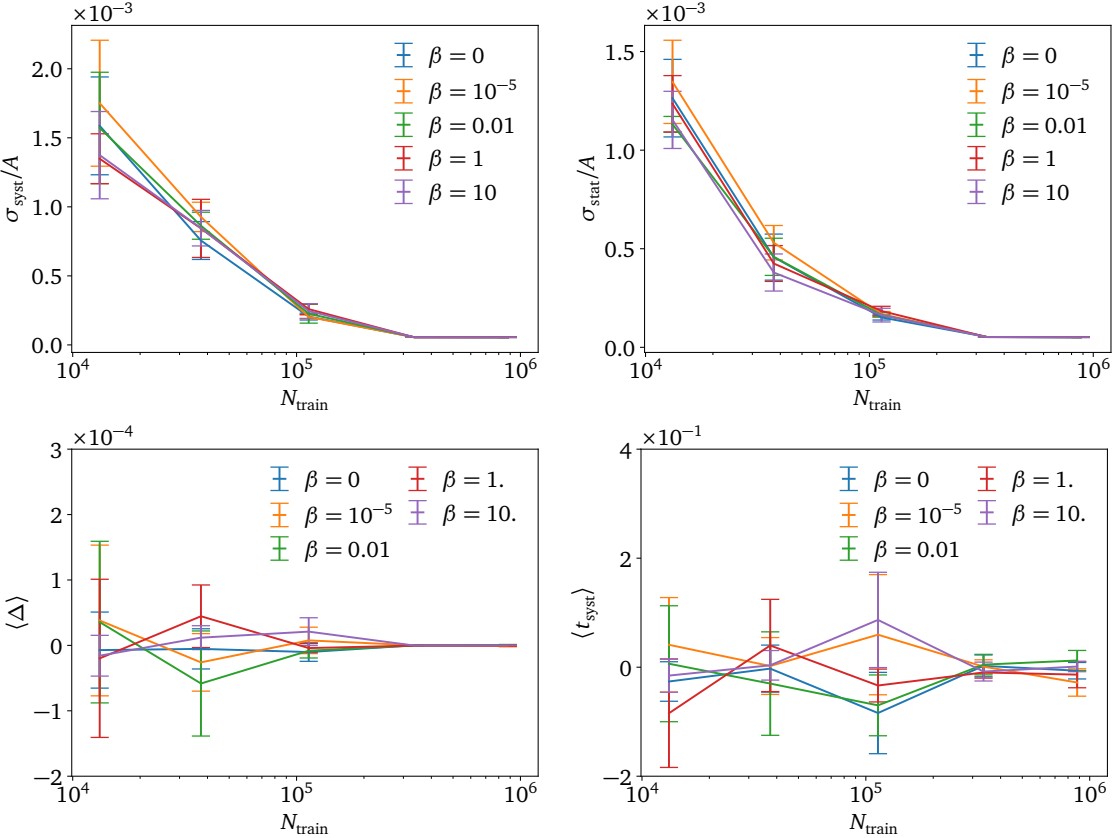

Figure 3: Relative uncertainty versus training dataset size for different kernel prefactors $\beta$. The plots show the relative systematic and statistical uncertainty, the mean accuracy $\langle \Delta \rangle$, and the mean systematic pull $\langle t_{\mathrm{syst}} \rangle$. The error bars are calculated based on five independent runs.

ensemble members. This spread of ensemble members is linked to $\beta$ as a repulsive prefactor in the loss function. However, this diversity in the spread of ensemble members is only connected to the relative size of the uncertainty estimation and not its spread. The spread of uncertainties and predictions is mainly influenced by the number of data points used for training. Including fewer points leads towards a more sensitive prediction based on the samples drawn, which can be described in analogy to artificial noise. With this, the prediction gets noisier and the relative uncertainty shows a larger spread compared to larger training data sets, where the training distribution converges more towards the actual data distribution.

## 3.2 Bias as the limitation of ensembling

Ensembles are often used to achieve accurate network predictions when individual network training lacks accuracy or stability. The implicit assumption is that a local ensemble mean provides an improved prediction, independent of the ensemble variance. As long as we are dominated by the training statistics, this is justified. For systematics, we need to ensure that there is no bias in the ensemble.

In the previous section, we observed a slight bias in the repulsive ensemble for small training datasets, independent of the repulsive kernel. The same bias can be observed for individually trained deterministic networks using a simple MLP architecture, confirming that the repulsive kernel is not related to the potential bias.

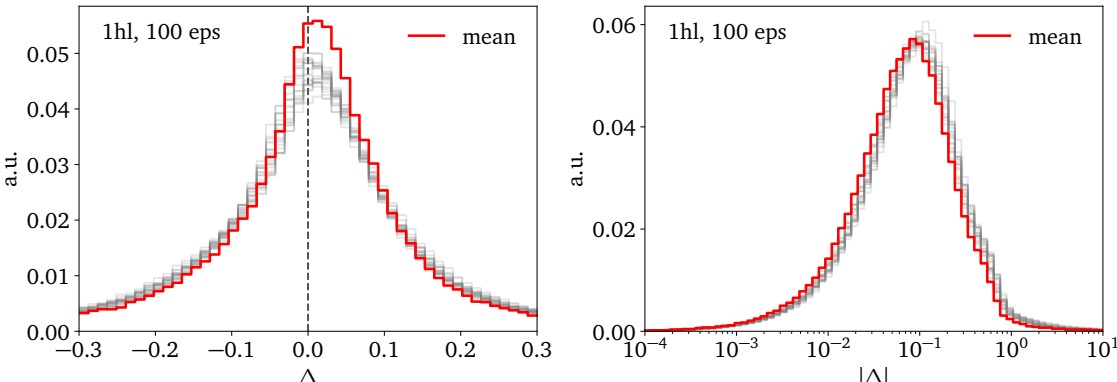

Figure 4: $\Delta$ distribution for a repulsive ensemble trained for 100 epochs with one hidden layer of dimension 32. The gray lines represent the individual ensemble members, while the red curve displays the mean over all members.

To test this effect in more detail, we employ different setups, varying the training length and the network depth in terms of the number of layers and dimensionality. The number of training points is fixed to the relative large number $N_{\text{train}} = 777102$, or roughly 70% of the full set. Tab. 1 shows the percentage of phase space points with a negative or a positive bias, as well as the sum over the bias for both sides. An ideal network would return $\Delta = 0$ everywhere. For $\Delta < 0$ the network underestimates $A_{\text{NN}}$ relative to $A_{\text{true}}$, for $\Delta > 0$ it overestimates it. A calibrated network should give equal numbers of points with negative and positive shifts, and the overall sum should lead to 0.

As the best-performing setup in Tab. 1, an ensemble with three layers and 128 dimensions, trained for 1000 epochs, shows almost no bias. If we reduce the number of layers to one and the dimensionality to 32, but keep the number of training epochs fixed, we observe a bias towards more positive $\Delta$. There, the network overestimates the amplitudes. Also, reducing the training time from 1000 epochs by a factor of 10 to 100 epochs has no significant impact on the bias. Additionally, considering all ensemble members and averaging them versus only taking single members into account does not influence the bias. This highlights the negligible impact of the ensemble compared to a single deterministic network in terms of the bias.

Figure 4 shows the distribution of the relative precision $\Delta$ (left) and its absolute value $|\Delta|$ (right) for the smallest training setup, using a single hidden layer with 32 units trained for 100 epochs. In the left panel, the dashed vertical line marks the unbiased optimum at $\Delta = 0$. The peak of the distribution is shifted towards positive values, indicating that the network systematically overestimates the amplitude. This bias appears both in the individual ensemble members (gray curves) and in their mean (red curve). In contrast, the right panel shows that

| ensemble configuration | mean | neg. $\Delta$ | sum neg. $\Delta$ | pos. $\Delta$ | sum pos. $\Delta$ |
|---|---|---|---|---|---|
| 1 hl, 32 dim, 1000 epochs | all | 45.40% | -9235.19 | 54.60% | 32455.27 |
| 1 hl, 32 dim, 100 epochs | all | 44.48% | -11565.01 | 55.52% | 32080.50 |
| 1 hl, 32 dim, 100 epochs | single | 44.48% | -11565.01 | 55.52% | 32080.50 |
| 3 hl, 128 dim, 1000 epochs | all | 49.85% | -221.02 | 50.15% | 219.62 |

Table 1: Relative accuracy $\Delta$ for every predicted amplitude, separated into positive and negative contributions varying the expressivity of the network. 'all' indicates the mean over all ensemble members, 'single' only for a single member.

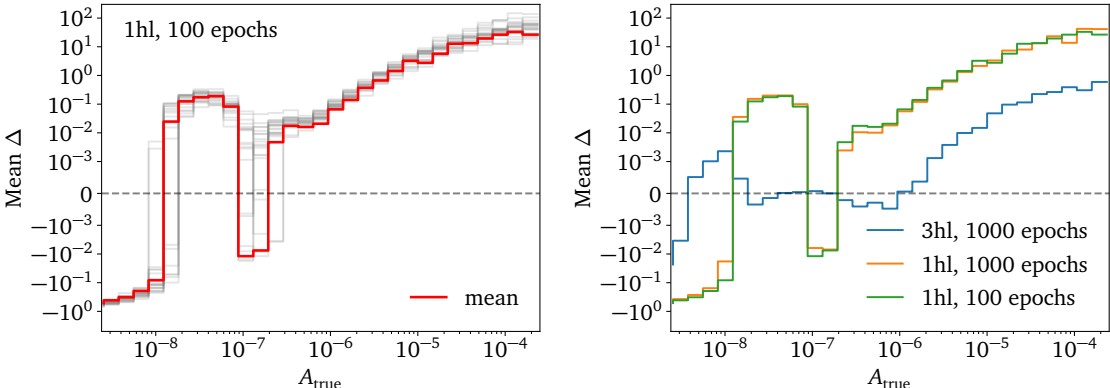

Figure 5: Mean value for $\Delta$ calculated bin-wise for the true amplitudes $A_{\text{true}}$. Left: Comparing a full ensemble with its single member contribution. Right: Showing different network sizes and configurations in training length.

the absolute relative precision $|\Delta|$ is essentially the same for both individual members and the mean prediction. We conclude that while the network exhibits a small positive bias, it still maintains its predictive precision.

To further investigate the origin of the bias, we analyze how it depends on the size of the true amplitude $A_{\text{true}}$. For this purpose, we bin the amplitude space and compute the mean relative precision $\langle \Delta \rangle$ in each bin, as shown in Fig. 5. The left panel shows the results for the small ensemble setup already used in Fig. 4. Both the individual ensemble members (gray curves) and the mean prediction (red curve) exhibit the same behavior: very small amplitudes are slightly underestimated, intermediate amplitudes up to $A_{\text{true}} \sim 10^{-6}$ are mildly overestimated, and large amplitudes are strongly overestimated, with a bias reaching $\langle \Delta \rangle \lesssim 100$. The right panel compares different network configurations. Increasing the training time from 100 to 1000 epochs does not affect the bias, indicating that it is not due to insufficient training. In contrast, enlarging the network capacity (three hidden layers with 128 units) substantially reduces the bias at large amplitudes by up to two orders of magnitude. This demonstrates that the bias is primarily a consequence of limited model expressivity. The effect is most pronounced at large amplitudes, where the training data become sparse (see Fig. 2); in this regime, a small residual bias persists even for larger networks.

Overall, small network setups yield biased deterministic predictions. This bias cannot be removed by using an ensemble: as shown in Fig. 4, it is present in each individual ensemble member and does not cancel when averaging over them. Moreover, extending the training time does not mitigate the effect. Only increasing the network expressivity reduces the bias significantly. Importantly, this behavior is not an artifact of fitting in log-amplitude space and transforming back: given the extremely small typical relative deviations ($\langle |\Delta| \rangle \sim 10^{-5}$), the exponential mapping is effectively linear in the relevant regime. We have explicitly verified that the same bias persists when analyzing the predictions directly in log space, confirming that its origin lies in limited model expressivity rather than in the post-processing transformation.

## 3.3 Systematics from repulsive ensembles

The occurrence of biases has an immediate effect on the calibration of the systematic uncertainty of an ensemble.

The naive heteroscedastic loss for $N_{\text{ens}}$ repulsive ensemble members trained on batches with $B$ was given by Eq.(23), where $\bar{A}(x, \theta_i)$ and $\sigma(x, \theta_i)$ are the two outputs for each ensemble

member. The combined predictions from the ensemble are

$$A_{\text{NN}}(x) = \frac{1}{N_{\text{ens}}} \sum_{i=1}^{N_{\text{ens}}} \overline{A}(x, \theta_i) \qquad \sigma_{\text{syst}}^2(x) = \frac{1}{N_{\text{ens}}} \sum_{i=1}^{N_{\text{ens}}} \sigma^2(x, \theta_i) \,. \tag{25}$$

This implementation leads to the miscalibration of the systematic uncertainty if the dominant source of systematic uncertainties is not the noise of the data. To understand this, we consider our training data to be generated as

$$A_{\text{train}}(x) \sim \mathcal{N}(A_{\text{true}}, \sigma_{\text{train}}^2) \,, \tag{26}$$

where $\sigma_{\text{train}}$ encodes the noise of the data. In a simple NN learning process, we further assume that the mean prediction of each ensemble member does not perfectly reproduce the underlying truth. It deviates by a fixed bias term $\sigma_{\text{bias}}$ due to limited expressivity of the network, and fluctuates around it with an additional Gaussian uncertainty $\sigma_{\text{stat}}$ arising from imperfect training:

$$\overline{A}(x, \theta_i) \sim \mathcal{N}(A_{\text{true}} + \sigma_{\text{bias}}, \sigma_{\text{stat}}^2) \,. \tag{27}$$

The residuals entering the heteroscedastic loss are then given by the difference between the noisy training data and the imperfect network prediction,

$$A_{\text{train}}(x) - \overline{A}(x, \theta_i) \sim \mathcal{N}(\sigma_{\text{bias}}, \sigma_{\text{train}}^2 + \sigma_{\text{stat}}^2) \,. \tag{28}$$

Consequently, the predicted variance converges towards

$$\sigma^2(x, \theta_i) \simeq \sigma_{\text{bias}}^2 + \sigma_{\text{train}}^2 + \sigma_{\text{stat}}^2 \,, \tag{29}$$

showing explicitly that the heteroscedastic output absorbs both the data noise and the network-induced residual uncertainty. Here, we implicitly assume that the bias contribution can be represented as a Gaussian random effect since the heteroscedastic loss relies on a Gaussian likelihood. However, in practice, the loss absorbs all residual variance into $\sigma^2(x, \theta_i)$, regardless of its true underlying structure. Hence, if the bias exhibits non-Gaussian features — e.g. fixed offsets, skewness, or multi-modality — the Gaussian likelihood cannot properly capture these effects, and the resulting pulls become miscalibrated. Now, we can consider two limiting cases:

1. $\sigma_{\text{train}}^2 \gg \sigma_{\text{stat}}^2 + \sigma_{\text{bias}}^2$: The residuals between the training data and the NN predictions are dominated by the noise in the training labels,

$$A_{\text{train}}(x) - \overline{A}(x, \theta_i) \sim \mathcal{N}(0, \sigma_{\text{train}}^2) \,. \tag{30}$$

In this regime, the network still learns the underlying truth $A_{\text{true}}(x)$, but the residuals with respect to the noisy labels are driven by $\sigma_{\text{train}}^2$. Because this data-noise contribution is shared across all members, ensemble averaging does not reduce it, i.e.

$$A_{\text{train}}(x) - A_{\text{NN}}(x) \sim \mathcal{N}(0, \sigma_{\text{train}}^2) \,. \tag{31}$$

Moreover, each $\sigma(x, \theta_i)$ predicted by the heteroscedastic loss converges to $\sigma_{\text{train}}$. Consequently, the averaged ensemble output for $\sigma_{\text{syst}}$ from Eq.(25) correctly approaches $\sigma_{\text{train}}$, and the systematic uncertainty is well calibrated, as shown numerically in Ref. [23].

2. $\sigma_{\text{train}}^2 \ll \sigma_{\text{stat}}^2 + \sigma_{\text{bias}}^2$: In this regime, the residuals are dominated by model-induced uncertainties,

$$A_{\text{train}}(x) - \overline{A}(x, \theta_i) \sim \mathcal{N}(\sigma_{\text{bias}}, \sigma_{\text{stat}}^2) \,. \tag{32}$$

In an ideal scenario, the statistical part would be captured entirely by the spread of ensemble predictions, as defined in Eq.(3). In practice, however, the heteroscedastic loss tends to also absorb some fraction of the statistical uncertainty and the variance predictions are given by

$$\sigma^2(x, \theta_i) \approx \sigma_{\text{bias}}^2 + \epsilon_{\text{het}} \sigma_{\text{stat}}^2 \qquad \text{with} \quad 0 \leq \epsilon_{\text{het}} < 1 . \tag{33}$$

If the ensemble members behave approximately as independent Gaussian estimators, the residual of the ensemble mean is distributed as

$$A_{\text{train}}(x) - A_{\text{NN}}(x) \sim \mathcal{N}(\sigma_{\text{bias}}, \sigma_{\text{mean}}^2) \qquad \text{with} \quad \sigma_{\text{mean}}^2 = \frac{\epsilon_{\text{het}} \sigma_{\text{stat}}^2}{N_{\text{ens}}} . \tag{34}$$

In contrast, the averaged ensemble output from Eq.(25) remains at $\sigma_{\text{syst}}^2 = \sigma_{\text{bias}}^2 + \epsilon_{\text{het}} \sigma_{\text{stat}}^2$, which does not follow the correct scaling with $N_{\text{ens}}$. This mismatch leads to a miscalibration of the systematic uncertainty for $N_{\text{ens}} > 1$, in agreement with Ref. [23].

As an alternative to a simple average, one may combine the outputs of the ensemble members with inverse-variance weighting,

$$A_{\text{NN}}(x) = \sigma_{\text{syst}}^2(x) \sum_{i=1}^{N_{\text{ens}}} \frac{\overline{A}(x, \theta_i)}{\sigma^2(x, \theta_i)} \qquad \text{with} \qquad \sigma_{\text{syst}}^2(x) = \left( \sum_{i=1}^{N_{\text{ens}}} \frac{1}{\sigma^2(x, \theta_i)} \right)^{-1} . \tag{35}$$

In principle, this approach incorporates the expected $1/\sqrt{N_{\text{ens}}}$ scaling of the statistical component. However, as discussed in Sec. 3.2 the ensemble members are not unbiased estimators of $A_{\text{true}}(x)$ for the $gg \to \gamma\gamma g$ process. Consequently, even with weighted averaging, the systematic uncertainties remain miscalibrated in either the network-error–dominated or the data-noise–dominated regime.

**Globally learned systematic uncertainty**

A possible solution is to first train the ensemble with the original loss of Eq.(23). Then, in a second step, we train an additional NN with parameters $\phi$ to directly predict a global systematic uncertainty $\sigma_{\text{syst}}^2(x, \phi)$ within the loss

$$\mathcal{L}_\sigma = \frac{1}{B} \sum_{b=1}^{B} \left[ \frac{|A_{\text{train}}(x_b) - A_{\text{NN}}(x_b)|^2}{2\sigma_{\text{syst}}^2(x_b, \phi)} + \log \sigma_{\text{syst}}(x_b, \phi) \right] , \tag{36}$$

where $A_{\text{NN}}$ is the averaged output of the ensemble trained in the first step, as defined in Eq.(35). The statistical uncertainty is still determined from the variance of the ensemble members trained in the first step. In practice, we can also combine the normal repulsive ensemble loss and the $\mathcal{L}_\sigma$ into one loss

$$\mathcal{L} = \mathcal{L}_{\text{RE}} + \lambda_\sigma \mathcal{L}_\sigma , \tag{37}$$

where we typically choose $\lambda_\sigma = N_{\text{ens}}$ to balance both loss terms. We find that the simultaneous training of the repulsive ensemble and the systematic uncertainty for the ensemble mean is useful to prevent mode collapse in the training of $\sigma_{\text{syst}}^2(x, \phi)$.

The accuracy for the $\gamma\gamma g$ amplitude regression is shown in the left panel of Fig. 6 for the zero noise case. Here, each member is a simple MLP with invariants and four vectors as input. With an increasing number of ensemble members, the accuracy of the ensemble improves. As expected, using the separately trained $\sigma_{\text{syst}}$ does not affect the accuracy of the ensemble.

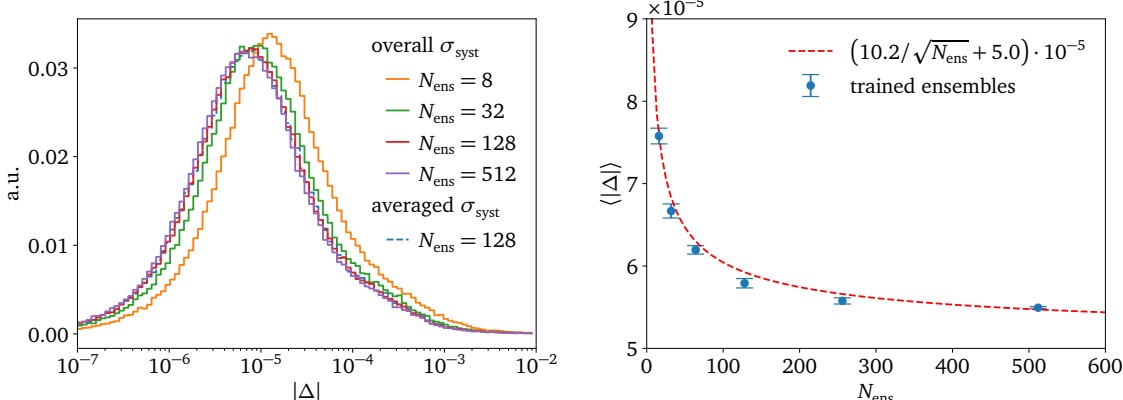

Figure 6: Relative accuracy $|\Delta|$ comparing the overall $\sigma_{\text{syst}}$ for different ensemble sizes $N_{\text{ens}}$ (solid lines) with the averaged $\sigma_{\text{syst}}$ for $N_{\text{ens}} = 128$ (blue dashed). The two $N_{\text{ens}} = 128$ results coincide within plotting resolution, causing the dashed curve to be hidden behind the solid red one. Right: Mean relative accuracy as a function of the number of ensemble members. The error bars indicate the standard deviation computed over five different runs.

Moreover, the right panel of Fig. 6 shows the mean relative accuracy — averaged over the test dataset — as a function of the ensemble size. The accuracy improves with $N_{\text{ens}}$ but levels off at around $\sim 5 \cdot 10^{-5}$. The behavior is well described by an inverse square-root scaling with a constant offset. The $1/\sqrt{N_{\text{ens}}}$ term demonstrates that the ensemble members act as approximately independent estimators, so that the statistical component decreases with ensemble size. In contrast, the constant offset corresponds to the irreducible bias, which cannot be reduced by ensembling. This numerical result directly confirms the conceptual decomposition discussed above: the ensemble reduces the statistical part as expected, but a bias floor remains.

Next, we discuss in Fig. 7, the learned systematic uncertainties of the ensemble. For the zero-noise case shown in the left panel, it is clearly visible that using the averaged systematic uncertainties from the ensemble members leads to overestimated uncertainties. We also checked that scaling the average by $1/\sqrt{N_{\text{ens}}}$ — or equivalently using the weighted average

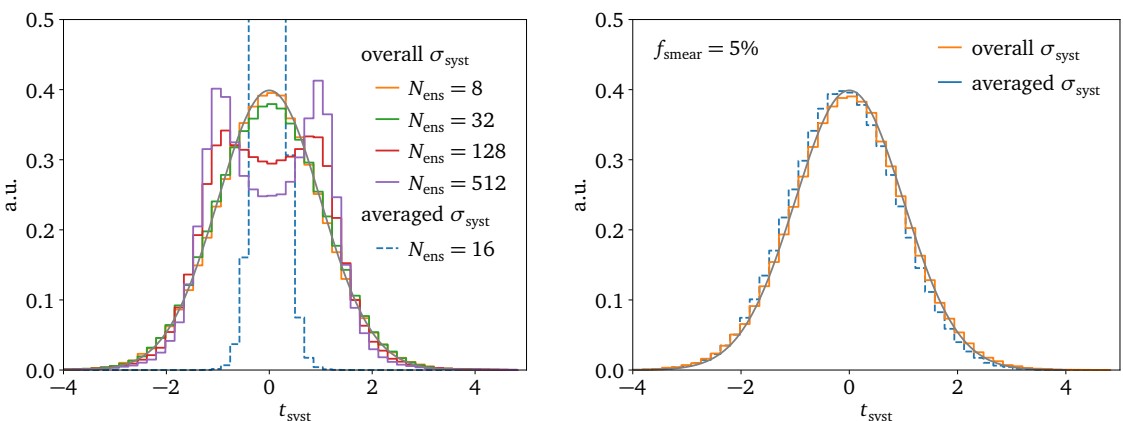

Figure 7: Left: systematic pull comparing the implementation with individual $\sigma_i$ for each ensemble member and the global $\sigma$ of Eq.(36) for different number of ensemble members. Right: same comparison using smeared data.

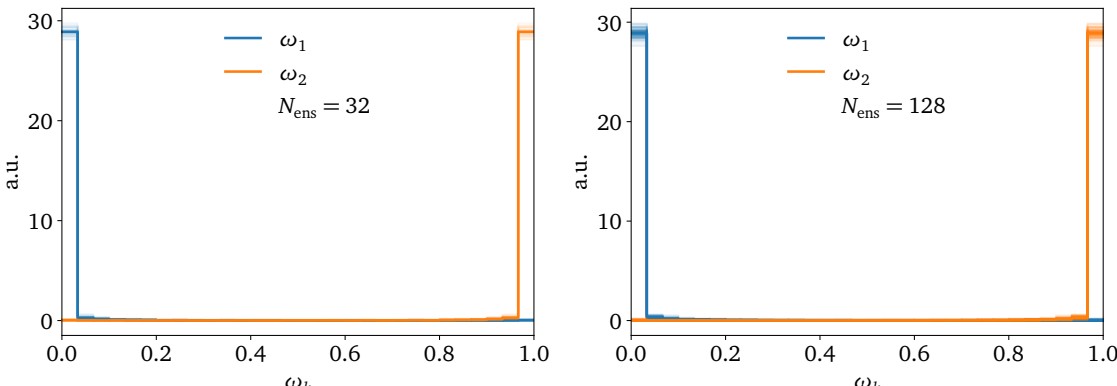

Figure 8: Distribution of the weights $\omega_k$ for a GMM with two modes. The orange line represents the distribution of $\omega_1$ for the first mode, and the blue line represents the distribution of $\omega_2$, corresponding to the second mode. The bold lines show the mean of $\omega_1$ or $\omega_2$ over all ensemble members $N_{\text{ens}}$.

of Eq.(35) — underestimates the uncertainties. Using instead a separately trained global $\sigma_{\text{syst}}$ drastically improves the calibration. For $N_{\text{ens}} \gtrsim 100$, however, two peaks start to appear roughly at $t_{\text{syst}} \sim \pm 1$. For this high number of ensemble members, the noisy part of the prediction is reduced to a level at which the biases of the learned prediction become clearly visible. If the noise in the amplitude prediction is low enough, the NN predicting $\sigma_{\text{syst}}(x)$ can directly fit $|A_{\text{NN}}(x) - A_{\text{train}}(x)|$, which is the actual minium of the heteroscedastic loss — see also App. D.3 of Ref. [65]. Consequently, for larger ensembles with low noise, $t_{\text{syst}} = (A_{\text{NN}}(x) - A_{\text{train}}(x))/\sigma_{\text{syst}}(x) \sim \pm 1$ explaining the appearance of the peaks. This signals that the uncertainty is not Gaussian distributed anymore.

In the case of noisy data, as shown in the right panel of Fig. 7, both the averaged and the separately learned global $\sigma_{\text{syst}}$ lead to well-calibrated uncertainties. This behavior is fully consistent with the data-noise–dominated scenario discussed above, where the irreducible noise contribution is shared across all ensemble members and is therefore not reduced by averaging.

**Gaussian mixture model**

Instead of assuming a Gaussian likelihood, we can also employ a Gaussian mixture model (GMM), as introduced in Sec. 2.2. For this, we use a repulsive ensemble architecture, as in Sec. 3.3, consisting of 32 or 128 ensemble members and train them with a GMM likelihood featuring two modes, i.e. $K = 2$ in Eq.(19). In contrast to the Gaussian case, where the mean $\bar{A}(x, \theta)$ enters Eq.(25), we always use the MAP estimate $A_{\text{GMM}}^{\text{MAP}}(x, \theta)$ for each ensemble member, since this provides a more stable prediction in the multi-modal case, as discussed below Eq.(18). The ensemble combination therefore becomes

$$A_{\text{NN}}(x) = \frac{1}{N_{\text{ens}}} \sum_{i=1}^{N_{\text{ens}}} A_{\text{GMM}}^{\text{MAP}}(x, \theta_i) \qquad \sigma_{\text{syst}}^2(x) = \frac{1}{N_{\text{ens}}} \sum_{i=1}^{N_{\text{ens}}} \sigma_{\text{GMM}}^2(x, \theta_i), \qquad (38)$$

where $\sigma_{\text{GMM}}^2$ and $A_{\text{GMM}}^{\text{MAP}}$ are defined in Eqs.(17) and (18), respectively.

In Fig. 8, we show the distribution of the weights $\omega$ for the different modes. On the left-hand side, we show the results for a repulsive GMM model with 32 ensemble members; on the right-hand side, for 128 ensemble members. The blue lines indicate the weight distribution for one of the two Gaussian modes, and the orange lines indicate the second Gaussian mode. We

display the distribution of $\omega$ for every individual ensemble member. The bold line represents the mean of $\omega$ over all ensemble members, $N_{\mathrm{ens}}$. For both setups, we observe a clear separation of weights for both GMM modes. With this behavior, we conclude that only one Gaussian is necessary to model the likelihood, confirming our initial assumption of a Gaussian likelihood shape. This also confirms that the observed bias is indeed driven by model expressivity and not by a wrong likelihood assumption during the fit.

## 4 Evidential regression

While repulsive ensembles and BNNs encode the posteriors of the network parameters, evidential regression (ER) estimate statistical and systematic uncertainties without ensembling or sampling. Instead it places a prior over the likelihood describing parameters. To better understand this, we start by assuming again a Gaussian likelihood [67],

$$p(A|x,\lambda) = \mathcal{N}\left(A|\overline{A}(x), \sigma^2(x)\right), \tag{39}$$

where the likelihood parameters $\lambda = (\overline{A}, \sigma^2)$ are not treated as fixed network outputs anymore but as random variables. Similar to Eq.(1), we can then parametrize a predictive distribution $p(A|x)$ as

$$p(A|x) = \int d\lambda\, p(A|x,\lambda)\, p(\lambda|D_{\text{train}}) \approx \int d\lambda\, p(A|x,\lambda)\, p(\lambda|m). \tag{40}$$

In the last line, we approximate the intractable posterior $p(\lambda|D_{\text{train}})$ with $p(\lambda|m)$, parametrized by $m$. As $p(\lambda|m)$ serves as a higher-order distribution compared to the likelihood, it is denoted as *evidential distribution* and its parameters are called *evidential parameters*. Ideally, we want to choose this distribution such that it comes from the same distribution family as the posterior which makes them *conjugate distributions*. In this case, $p(\lambda|m)$ becomes the *conjugate prior* of the likelihood $p(A|x,\lambda)$. This implies that we can learn the parameters $m$ from the training data without changing the form of the distribution.

Given that we have chosen $p(A|x,\lambda)$ to be a Gaussian, the conjugate prior is mathematically given by the Normal-Inverse-Gamma (NIG) distribution

$$p(\lambda|m) = \frac{\beta^\alpha \sqrt{\nu}}{\Gamma(\alpha)\sqrt{2\pi\sigma^2}} \left(\frac{1}{\sigma^2}\right)^{\alpha+1} \exp\left(-\frac{2\beta + \nu(\gamma - \overline{A})^2}{2\sigma^2}\right), \tag{41}$$

where $\Gamma(\cdot)$ is the gamma function. The evidential parameters

$$m \equiv m(x,\theta) = \{\gamma, \nu, \alpha, \beta\}(x,\theta) \qquad \text{with} \qquad \nu > 0, \alpha > 1, \beta > 0. \tag{42}$$

are the outputs of a network with weights $\theta$, which is why it is denoted as *evidential regression*.

The conjugacy of the NIG distribution allows interpreting the parameters $m$ in the following way [79]: The sample mean $\gamma$ is estimated from $\nu$ observations. The corresponding variance is derived from $\alpha$ observations with mean $\gamma$ and the sum of squared deviations being $2\nu$. The combined NIG prior allows us to compute the mean amplitude and its two uncertainties as

$$A_{\text{NN}}(x) = \int dA\, A\, p(A|x)$$
$$= \int d\overline{A}\, d\sigma^2\, \overline{A}(x)\, p(\lambda|m) = \gamma,$$
$$\sigma_{\text{tot}}^2(x) = \int dA\, (A - A_{\text{NN}})^2\, p(A|x)$$
$$= \int d\overline{A}\, d\sigma^2\, \left(\sigma^2(x) + \left[\overline{A}(x) - A_{\text{NN}}(x)\right]^2\right) p(\lambda|m), \tag{43}$$

where the systematic and statistical uncertainty is thus given by

$$\sigma_{\text{syst}}^2(x) = \int d\overline{A}\, d\sigma^2\, \sigma^2(x)\, p(\lambda|m) = \frac{\beta}{\alpha - 1},$$
$$\sigma_{\text{stat}}^2(x) = \int d\overline{A}\, d\sigma^2\, \left[\overline{A}(x) - A_{\text{NN}}(x)\right]^2 p(\lambda|m) = \frac{\beta}{\nu(\alpha - 1)}. \tag{44}$$

Using the NIG distribution, the evidential likelihood for the amplitude can be obtained analytically. As shown in the appendix of Ref. [67], the evidential likelihood is given by

$$
\begin{aligned}
p(A|x,m) &= \int \mathrm{d}\lambda\, p(A|x,\lambda)\, p(\lambda|m) \\
&= \int_0^\infty \mathrm{d}\sigma^2 \int_{-\infty}^\infty \mathrm{d}\bar{A}\, p(A|\bar{A},\sigma^2)\, p(\bar{A},\sigma^2|\gamma,\nu,\alpha,\beta) \\
&= \int_0^\infty \mathrm{d}\sigma^2 \int_{-\infty}^\infty \mathrm{d}\bar{A} \left[ \frac{1}{\sqrt{2\pi\sigma^2}} \exp\left( -\frac{(A-\bar{A})^2}{2\sigma^2} \right) \right] \\
&\quad \times \left[ \frac{\beta^\alpha \sqrt{\nu}}{\Gamma(\alpha)\sqrt{2\pi\sigma^2}} \left( \frac{1}{\sigma^2} \right)^{\alpha+1} \exp\left( -\frac{2\beta + \nu(\gamma-\bar{A})^2}{2\sigma^2} \right) \right] \\
&= \int_0^\infty \mathrm{d}\sigma^2 \frac{\beta^\alpha \sigma^{-3-2\alpha}}{\sqrt{2\pi}\sqrt{1+1/\nu}\,\Gamma(\alpha)} \exp\left( -\frac{2\beta + \frac{\nu(A-\gamma)^2}{1+\nu}}{2\sigma^2} \right) \\
&= \frac{\Gamma(\alpha+1/2)}{\Gamma(\alpha)} \sqrt{\frac{\nu}{\pi}} \left( 2\beta(1+\nu) \right)^\alpha \left( \nu(A-\gamma)^2 + 2\beta(1+\nu) \right)^{-\alpha-\frac{1}{2}} \\
&= \mathrm{St}\left( A \middle| \gamma, \frac{\beta(1+\nu)}{\nu\alpha}, 2\alpha \right).
\end{aligned}
\tag{45}
$$

where $\mathrm{St}(A|\mu_{\mathrm{St}}, \sigma^2_{\mathrm{St}}, \nu_{\mathrm{St}})$ denotes the Student-$t$ distribution with location $\mu_{\mathrm{St}}$, scale $\sigma^2_{\mathrm{St}}$ and $\nu_{\mathrm{St}}$ degrees of freedom. For simplicity, we suppressed the dependence on the phase space point $x$ in this derivation. The log-likelihood loss directly follows as

$$
\begin{aligned}
\mathcal{L}_{\mathrm{St}} &= -\sum_i \log p(A|x_i, m_i) \\
&= -\sum_i \log\left( \mathrm{St}\left( A \middle| \gamma_i, \frac{\beta_i(1+\nu_i)}{\nu_i\alpha_i}, 2\alpha_i \right) \right) \\
&= \sum_i \left[ \left( \alpha_i + \frac{1}{2} \right) \log\left[ \nu_i (A_{\mathrm{train}}(x_i) - \gamma_i)^2 + \Omega_i \right] + \log\frac{\Gamma(\alpha_i)}{\Gamma\left(\alpha_i + \frac{1}{2}\right)} + \frac{1}{2}\log\frac{\pi}{\nu_i} - \alpha_i \log\Omega_i \right] \\
&\quad \text{with} \quad \Omega_i = 2\beta_i(1+\nu_i).
\end{aligned}
\tag{46}
$$

Since the Student-$t$ distribution depends on only three effective parameters, minimizing the likelihood alone does not uniquely constrain the four outputs $(\gamma, \nu, \alpha, \beta)$. This leads to a degeneracy in the learned evidential parameters. To address this, we introduce the regularization loss [67]

$$
\mathcal{L}_R = \sum_i |A_{\mathrm{train}}(x_i) - \gamma_i| \cdot \Phi_i = \sum_i |A_{\mathrm{train}}(x_i) - \gamma_i| \cdot (2\nu_i + \alpha_i),
\tag{47}
$$

where $\Phi$ is the total evidence encoding the strength of belief in the predicted parameters. The regularization loss discourages the network from assigning high evidence to incorrect predictions. Specifically, when the predicted mean $\gamma$ deviates significantly from the target, the loss penalizes large values of the total evidence. Conversely, when the prediction is accurate, high evidence is not penalized. The combined evidential regression loss is

$$
\mathcal{L}_{\mathrm{ER}}^R = \mathcal{L}_{\mathrm{St}} + \lambda_R \mathcal{L}_R,
\tag{48}
$$

where $\lambda_R$ is a tunable hyperparameter. We set $\lambda_R = 0.01$ by default.

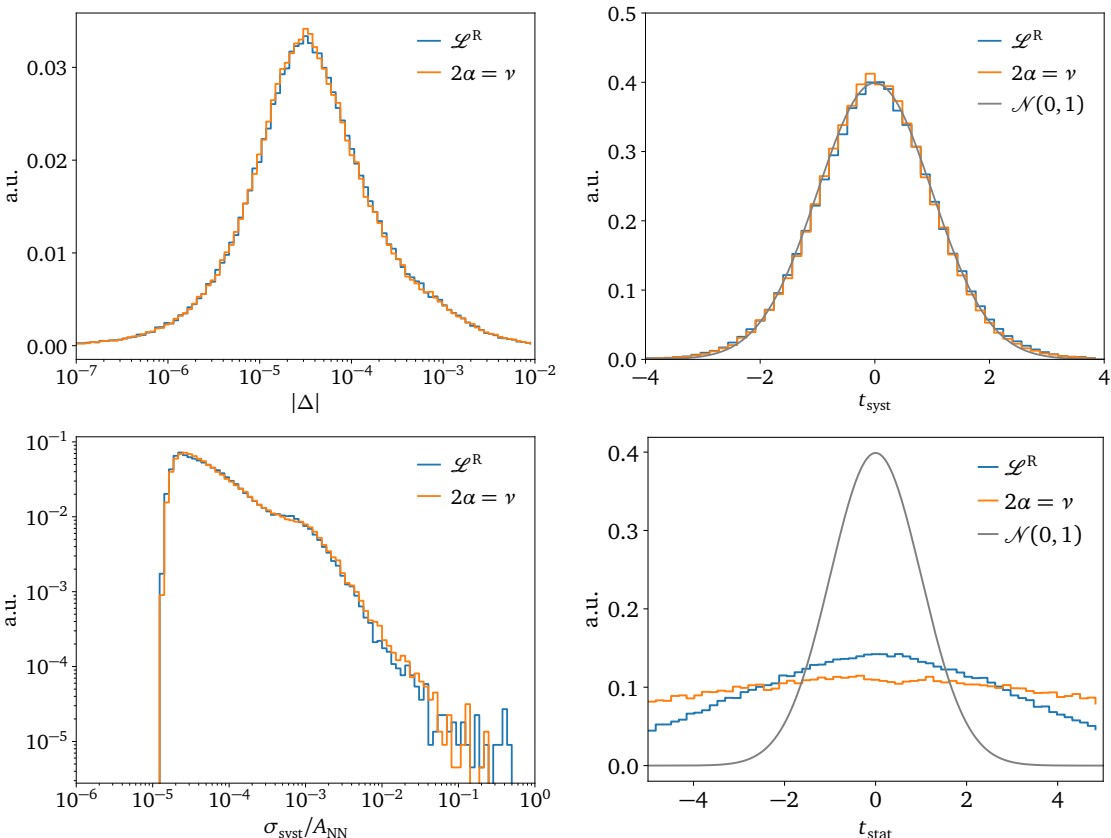

Figure 9: Evidential regression results for unsmeared $gg \to \gamma\gamma g$ dataset. The results with regularization loss are compared to the results setting $2\alpha = \nu$.

An alternative to using the regularization loss is to constrain the evidential parameters directly. Following Ref. [68], we can fix the ratio between $\alpha$ and $\nu$ via

$$2\alpha = r\nu \, , \tag{49}$$

with a constant $r$. One can show that in the limit of vanishing statistical uncertainty — corresponding to $\nu \to \infty$ — the predictive likelihood should converge to a Gaussian. This requires $\alpha \to \infty$ as well, which is automatically ensured by this constraint. In this case, the loss is given by the same negative log-likelihood as in Eq.(46), with $\nu$ replaced by $2\alpha/r$,

$$\mathcal{L}_{\mathrm{ER}} = \sum_i \left[ \left( \alpha_i + \frac{1}{2} \right) \log \left[ \frac{2\alpha_i}{r} \left( A_{\mathrm{train}}(x_i) - \gamma_i \right)^2 + \Omega_i \right] \right.$$
$$\left. + \log \frac{\Gamma(\alpha_i)}{\Gamma\left( \alpha_i + \frac{1}{2} \right)} + \frac{1}{2} \log \frac{\pi r}{2\alpha_i} - \alpha_i \log \Omega_i \right] \quad \text{with} \quad \Omega_i = 2\beta_i (1 + (2\alpha_i/r)) \, . \tag{50}$$

If not mentioned otherwise, we choose $r = 1$.

**Performance**

We now apply evidential regression to the $gg \to \gamma\gamma g$ dataset and compare two approaches for breaking the degeneracy in the base loss. The results are summarized in Fig. 9. The upper left panel shows the distribution of the absolute relative deviation from the true amplitudes. Both approaches achieve nearly identical precision with $\langle |\Delta| \rangle \sim 3 \cdot 10^{-5}$, comparable to the accuracy

obtained with a deterministic or Bayesian neural network [23]. The systematic calibration curves in the upper right panel confirm that the uncertainties are well calibrated. The lower left panel further shows the ratio $\sigma_{\mathrm{syst}}/A_{\mathrm{NN}}$, which is in close agreement with the results reported in Ref. [23]. We also find the predicted statistical uncertainties to be significantly smaller than the systematic ones. This is reflected in the broad statistical pull distributions in the lower right panel and mirrors the behavior observed for BNNs in Ref. [23]. Finally, enforcing the constraint $2\alpha = \nu$ yields slightly smaller statistical uncertainties than the alternative with an additional regularization loss.

Although not shown here, we also verified that applying a universal Gaussian smearing to the entire dataset yields well-calibrated uncertainties, consistent with the findings of Ref. [23] for repulsive ensembles and Bayesian NNs.

**Gaussian mixture model**

Instead of using a GMM only for a repulsive ensemble, one can, in principle, also integrate the GMM into the evidential regression setup. Based on the results from the repulsive ensemble GMM and the evidential regression network in Fig. 9, it is well motivated to assume that the evidential GMM would yield a similar outcome. We therefore only outline the conceptual setup here and refer to Ref. [80] for related work.

We start by replacing the single Gaussian in Eq.(39) with a $K$-component GMM,

$$p_{\mathrm{GMM}}(A|x,\{\lambda_k\}) = \sum_{k=1}^{K} \omega_k(x) \mathcal{N}\big(A\big|\bar{A}_k(x),\sigma_k^2(x)\big), \qquad \sum_{k=1}^{K} \omega_k(x) = 1\,, \tag{51}$$

where each component $\lambda_k = \{\bar{A}_k,\sigma_k\}$ has its own conjugate NIG prior

$$p(\lambda_k|m_k) = \frac{\beta_k^{\alpha_k}\sqrt{\nu_k}}{\Gamma(\alpha_k)\sqrt{2\pi\sigma_k^2}} \left(\frac{1}{\sigma_k^2}\right)^{\alpha_k+1} \exp\left(-\frac{2\beta_k + \nu_k(\gamma_k - \bar{A}_k)^2}{2\sigma_k^2}\right)\,, \tag{52}$$

with evidential parameters $m_k = \{\gamma_k,\nu_k,\alpha_k,\beta_k\}$. Since the prior factorizes over components, the evidential likelihood becomes a weighted mixture of Student-$t$ distributions,

$$p_{\mathrm{GMM}}(A|x,\{m_k\}) = \sum_{k=1}^{K} \omega_k \, \mathrm{St}\left(A\Big|\gamma_k, \frac{\beta_k(1+\nu_k)}{\nu_k\alpha_k}, 2\alpha_k\right)\,. \tag{53}$$

The corresponding loss function is simply the negative log-likelihood of this mixture,

$$\begin{aligned}\mathcal{L}_{\mathrm{ER\text{-}GMM}} &= -\sum_i \log p_{\mathrm{GMM}}(A|x_i,\{m_k\}_i) \\ &= -\sum_i \log \sum_{k=1}^{K} \omega_{ki} \, \mathrm{St}\left(A\Big|\gamma_{ki}, \frac{\beta_{ki}(1+\nu_{ki})}{\nu_{ki}\alpha_{ki}}, 2\alpha_{ki}\right)\,.\end{aligned} \tag{54}$$

# 5 Localized learning challenges

So far, we have only considered the case of simple Gaussian noise in all of phase space. For realistic settings, numerical noise will, however, typically be localized in phase space — for instance, in regions where loop integrals become harder to compute. Thresholds are candidate structures because the amplitude can turn from a real to a complex number. In the following, we test whether our uncertainty estimation can reliably identify localized noise.

## 5.1 Flat-box threshold smearing

As a first test, we emulate numerical noise close to a threshold by applying Gaussian smearing with the relative strength $\epsilon$ to all amplitudes with an invariant mass of the final state particles $m_{\gamma\gamma g}$ close to the artificial threshold $m_{\text{thresh}}$ within a box of width $w$,

$$
A_{\text{train}}(x) = \begin{cases} \mathcal{N}(A_{\text{true}}(x), \epsilon A_{\text{true}}(x)) & \text{if } |m_{\gamma\gamma g}(x) - m_{\text{thresh}}| < w \\ A_{\text{true}} & \text{if } |m_{\gamma\gamma g}(x) - m_{\text{thresh}}| \geq w \end{cases} . \tag{55}
$$

For our numerical investigation, we use $m_{\text{thresh}} = 200\,\text{GeV}$ and vary $\epsilon$ and $w$.

**Repulsive ensemble**

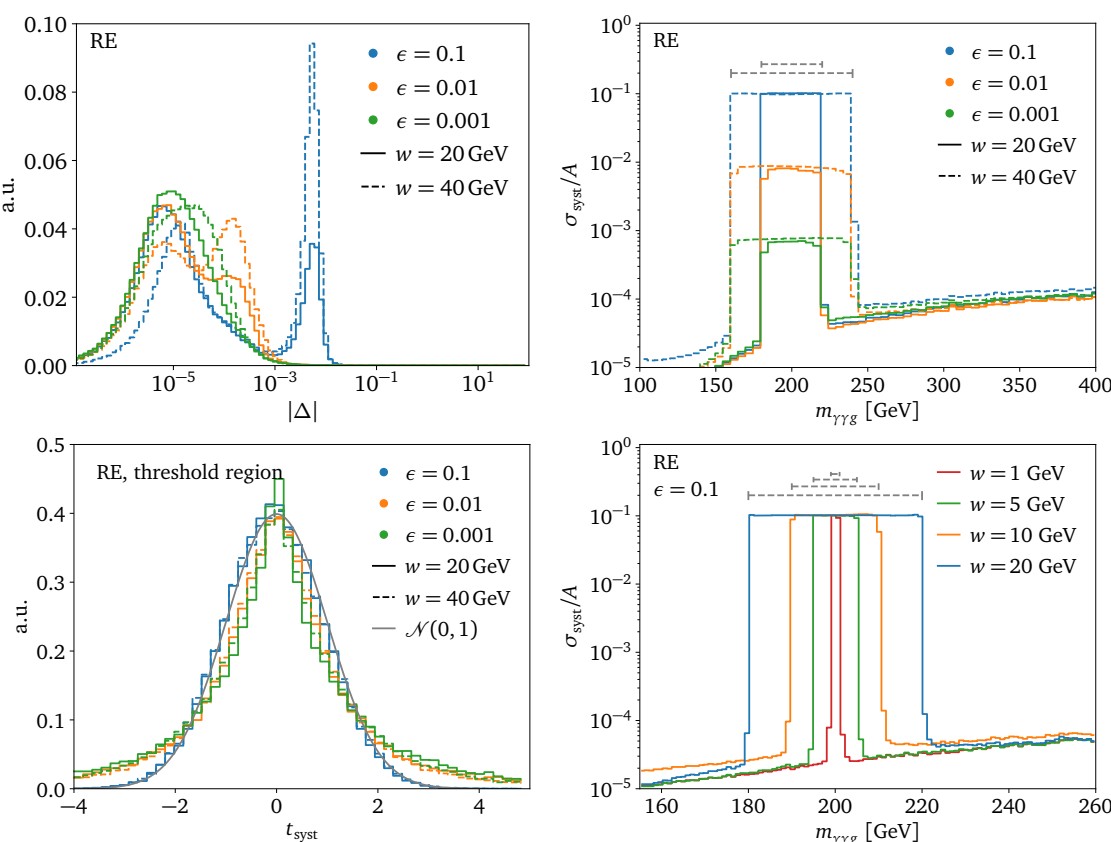

Figure 10: Upper left: $|\Delta|$ distributions for various choices of the threshold smearing strength $\epsilon$ and the threshold smearing window width $w$. Upper right: learned systematic error over learned amplitude as a function of $m_{\gamma\gamma g}$ for different choices of $w$ and $\epsilon$. The gray horizontal lines indicate the smearing window around $m_{\text{thresh}} = 200\,\text{GeV}$.

The results using repulsive ensembles are shown in Fig. 10. As expected, with induced noise the overall surrogate prediction gets worse compared to the noise-free case, as shown in the upper left panel. Especially for larger $\epsilon$ values and a larger box width, a second peak appears in the $|\Delta|$ distribution originating from the smeared amplitudes.

To investigate whether the learned systematic uncertainty correctly identifies the smeared phase-space region, we show in the upper right panel median $\sigma_{\text{syst}}/A$ binned in the invariant mass $m_{\gamma\gamma g}$. For a perfectly trained NN, $\sigma_{\text{syst}}/A$ should follow exactly the box form defined in Eq.(55). For all tested $\epsilon$ values, we indeed observe the NNs predictions to almost perfectly follow the expected box form. Outside of the smearing window, the curves fall back to the systematic uncertainty predicted without any applied smearing.

The systematic pull distributions — taking into account only events within the smearing box — are shown in the lower left panel. For $\epsilon = 0.1$, the systematic uncertainty is almost perfectly calibrated. For $\epsilon = 0.01$ and $\epsilon = 0.001$, the $t_{\text{syst}}$ distribution is not perfectly Gaussian anymore, which is likely due to the sharp edges in the smearing function.

In the lower right panel, we moreover test how the NN predictions behave for a shrinking smearing window. Up to $w = 1\,\text{GeV}$, the extracted noise level almost perfectly matches the expected form.

**Evidential regression**

For evidential regression, we now solely focus on the variant without the regularization loss, but imposing $2\alpha = \nu$, as shown in the results in Fig. 11. We find that the variant with regularization loss gives significantly worse results.

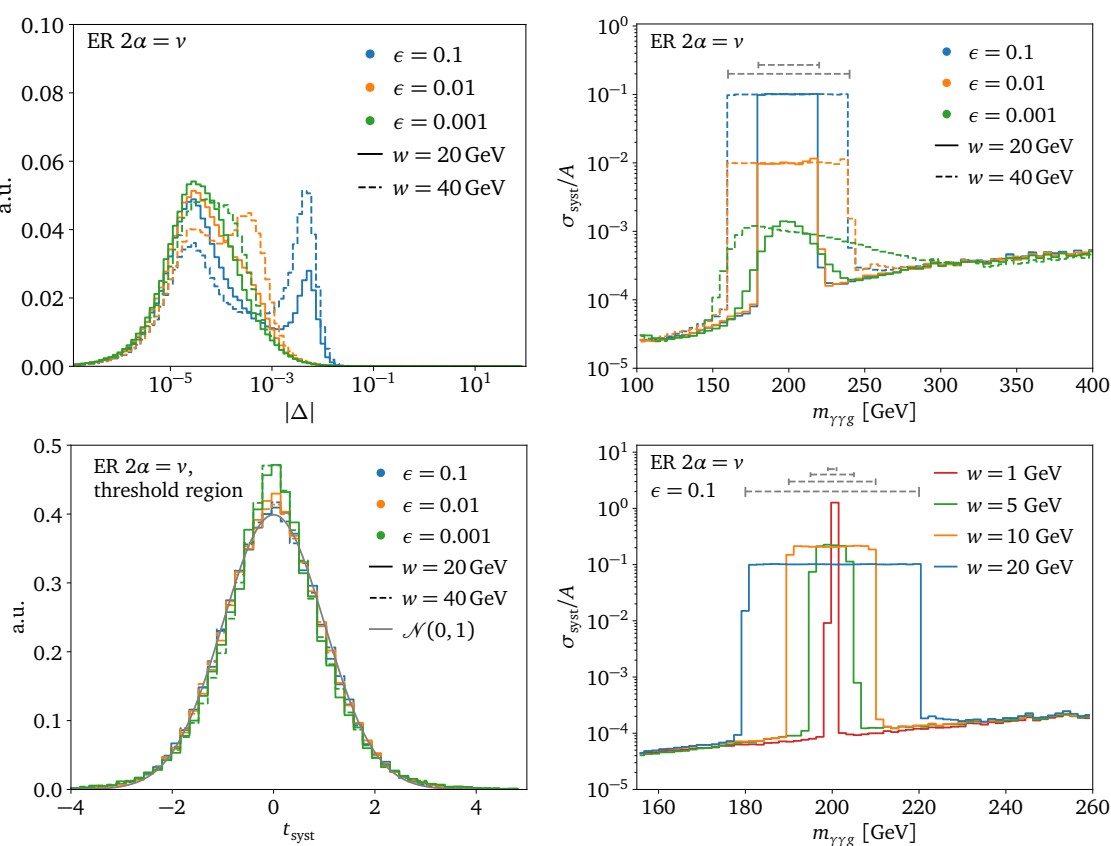

Figure 11: Evidential regression results for the box smearing approach.

The precision of the evidential regression is comparable to the repulsive ensemble, as visible in the upper left panel of Fig. 11. For $\epsilon = 0.001$, the evidential regression, however, is not able to predict the sharp edges of the flat box smearing, as shown in the upper right panel. For $\epsilon = 0.01$ and $\epsilon = 0.1$, the expected shapes are recovered. As visible in the lower left panel, the systematic uncertainties are well calibrated for all considered $\epsilon$ values. When lowering the window width $w$, see lower right panel, the evidential regression correctly captures the boundaries of the smearing box but struggles to extract the amount of smearing within the box for lower $w$ values.

**Bayesian neural network**

Additionally, we present the results for a BNN, in conjunction with the repulsive ensemble and evidential regression. As seen in the previous study [23], the BNN provides competitive results in terms of precision and uncertainty estimation compared to the repulsive ensemble. Fig. 12 shows that the BNN is as good as the repulsive ensemble in terms of the relative systematic uncertainty $\sigma_{\text{syst}}/A$ for various choices of $\epsilon$ and $w$. Additionally, the BNN provides a better-calibrated systematic uncertainty, as seen in the pull distribution in the lower left plot, which follows a Gaussian distribution. Comparing the precision of the amplitude estimation represented by $|\Delta|$, the BNN performs equally well as the evidential regression approach and thus is a well-justified and motivated approach to consider in these different localized learning challenges.

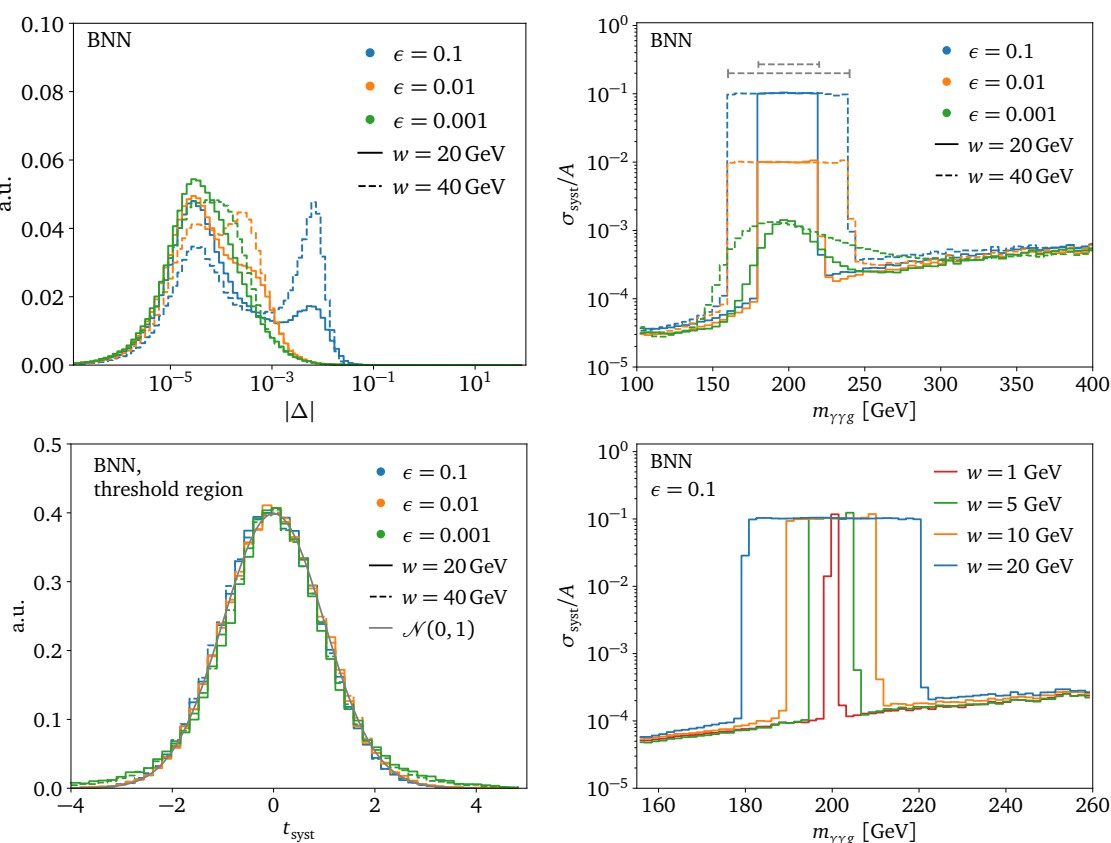

Figure 12: BNN results for the box smearing approach.

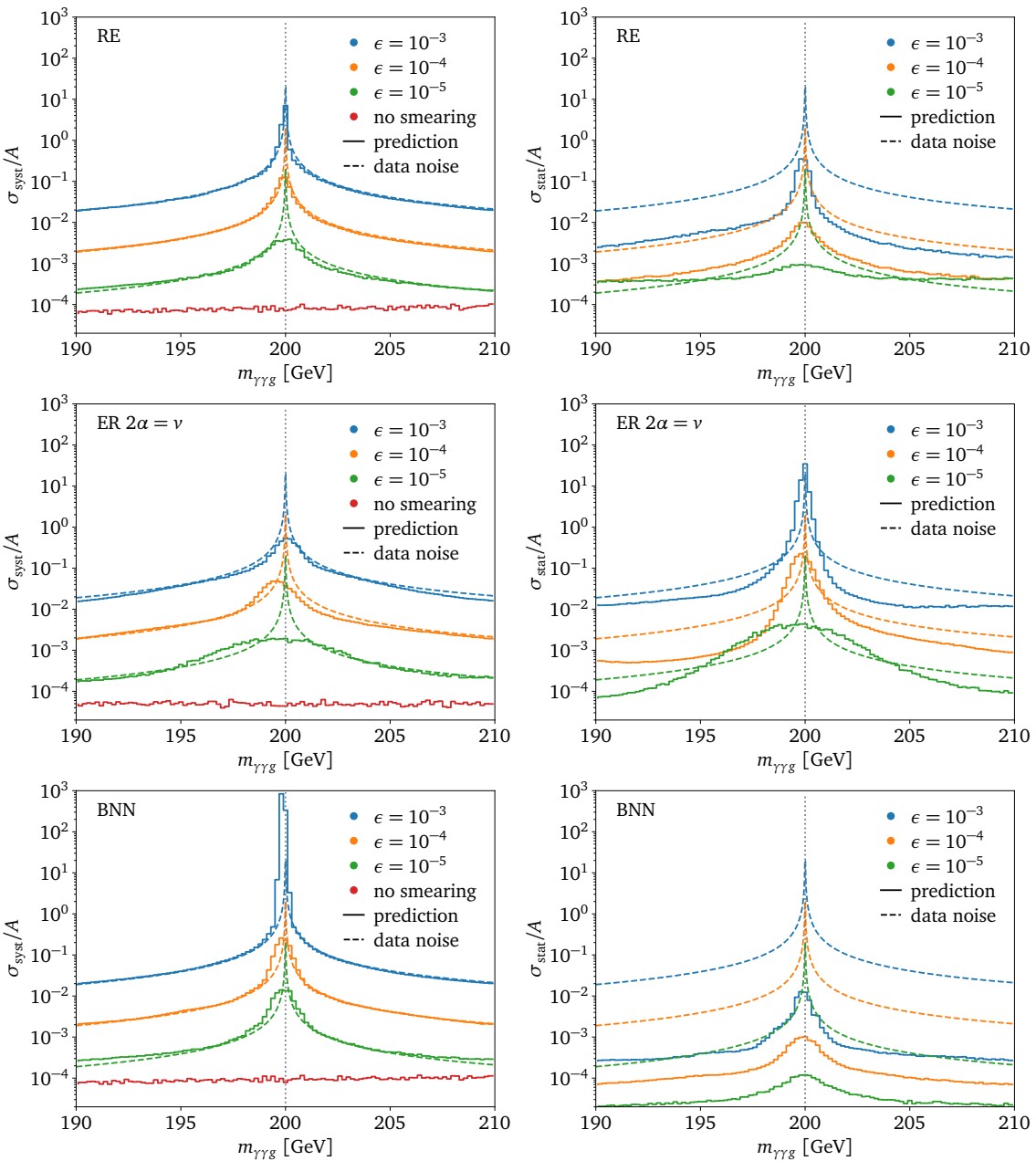

Figure 13: Left side: learned systematic uncertainty over learned amplitude as a function of $m_{\gamma\gamma g}$ for different choices of $\epsilon$ comparing repulsive ensemble results (upper row), evidential regression results (middle), and BNN results (lower row). Right side: learned statistical uncertainty over learned amplitude, displayed as described for the learned systematic uncertainty. The gray vertical line indicates the chosen threshold; the dashed lines, the expected behavior.

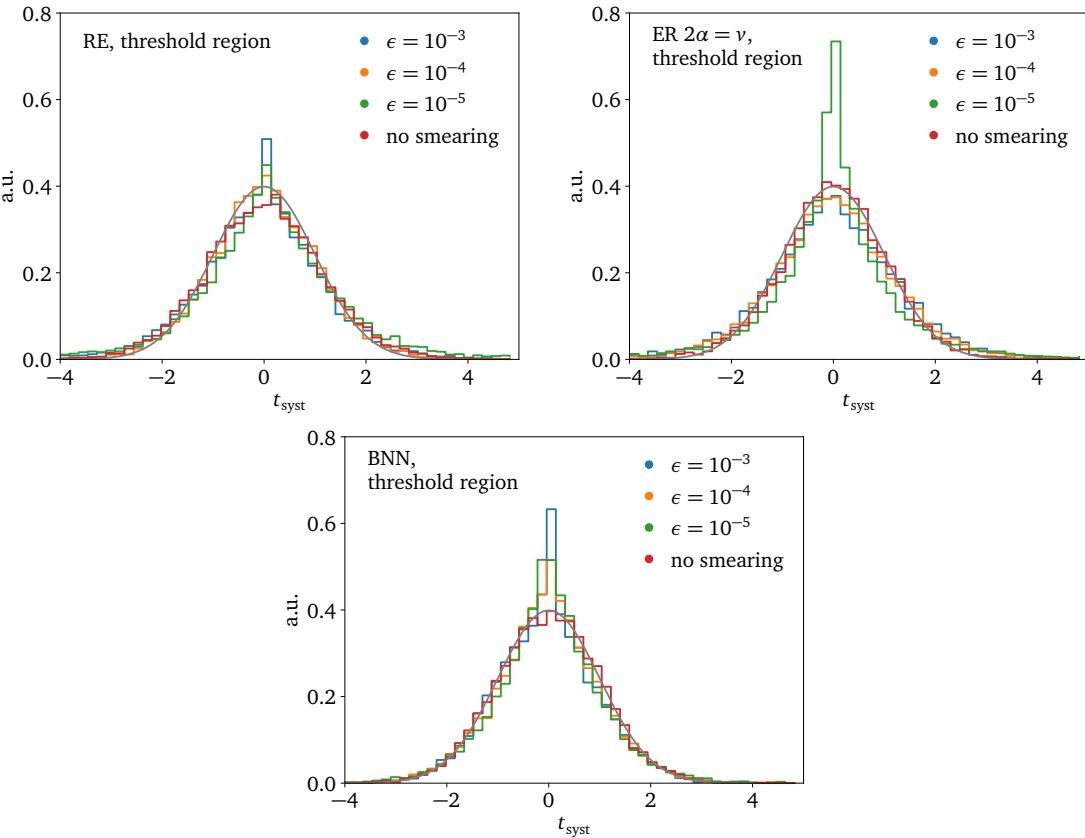

Figure 14: Results for the systematic pull distributions for events in the threshold region — 195 GeV $< m_{\gamma\gamma g} <$ 205 GeV — comparing repulsive ensemble results (upper left), evidential regression results (upper right), and BNN results (lower center).

## 5.2  Peaked threshold smearing

Next, we consider a more complicated noise profile for which the amount of noise increases close to the threshold value. Concretely, we use the smearing function

$$A_{\text{train}}(x) \sim \mathcal{N}\left(A; A_{\text{true}}(x), \frac{\epsilon\, m_{\text{thresh}}}{|m_{\gamma\gamma g}(x) - m_{\text{thresh}}|} A_{\text{true}}(x)\right), \qquad (56)$$

where we again choose $m_{\text{thresh}} = 200$ GeV.

In the left column of Fig. 13, the median $\sigma_{\text{syst}}/A$ for the binned $m_{\gamma\gamma g}$ distribution is shown for different values of $\epsilon$ using the repulsive ensemble (top), evidential regression (middle), and BNN approach (bottom). While evidential regression is able to capture the induced noise at least approximately, it struggles close to the threshold. In contrast, the repulsive ensemble is able to almost perfectly extract the noise. Only for $\epsilon = 10^{-5}$, the sharp increase in noise very close to the threshold is not perfectly captured. The BNN overestimates the relative systematic uncertainty for larger $\epsilon$, as shown for $\epsilon = 10^{-3}$. For the other choices of $\epsilon$, the BNN performs better than the evidential regression approach, but still has problems capturing the regions close to the threshold region correctly.

The right column shows the results for the relative statistical uncertainty $\sigma_{\text{stat}}/A$ using the same setup as in the left column, where the learned relative uncertainty (solid) is compared to its expected behavior from the underlying noise model (dashed). The predicted statistical uncertainty is consistently smaller than the expected noise level, showing that the networks

can learn an accurate amplitude prediction even in the presence of strong local smearing. This reflects a clear benefit of interpolation. Information from the surrounding clean regions stabilizes the prediction in the noisy region, enabling the networks to disentangle the smooth underlying amplitude from the localized noise. Comparing the three methods, we find that for a given $\epsilon$ the repulsive ensemble (top) yields a uniformly smaller $\sigma_{\text{stat}}/A$ than evidential regression (middle), with the BNN (bottom) providing the smallest relative statistical uncertainty $\sigma_{\text{stat}}/A$.

From a technical point of view, it is important not to use early stopping for small values of $\epsilon$. In this regime, only events very close to the threshold are significantly smeared. This can lead to outlier events being present in the training but not in the validation dataset or vice versa. As a result, the validation loss may temporarily increase while the training loss continues to decrease, without indicating actual overfitting.

In addition, Fig. 14 displays the systematic pull distributions for events of a $m_{\gamma\gamma g}$-range within 195 GeV and 205 GeV. Comparing these different pull distributions, the case without any smearing applied is well-calibrated for both the evidential regression and the BNN approach. In the case of smearing, the uncertainties are overestimated by all three approaches, particularly for evidential regression using $\epsilon = 10^{-5}$.

## 5.3 Threshold gap

Instead of locally smearing the true amplitude values, we now consider a different scenario: the absence of events in certain phase-space regions. In practice, such gaps can arise when the evaluation of the true amplitude fails, for instance near thresholds. Here, however, we deliberately enlarge the missing region to create a more severe case. While this setup is admittedly artificial, it serves as a valuable test for the statistical uncertainty prediction, which should increase significantly within regions lacking training data.

In particular, we remove events within $|m_{\gamma\gamma g}(x) - m_{\text{thresh}}| < 40$ GeV from the training and validation datasets. In contrast, the test dataset still contains events within the threshold region.

Here, we focus only on the repulsive ensemble and the BNN approach. The repulsive ensemble accuracy is shown in the left panel of Fig. 15 and the accuracy for the BNN is shown in the right panel. While the removal of events in the threshold region does affect the accuracy,

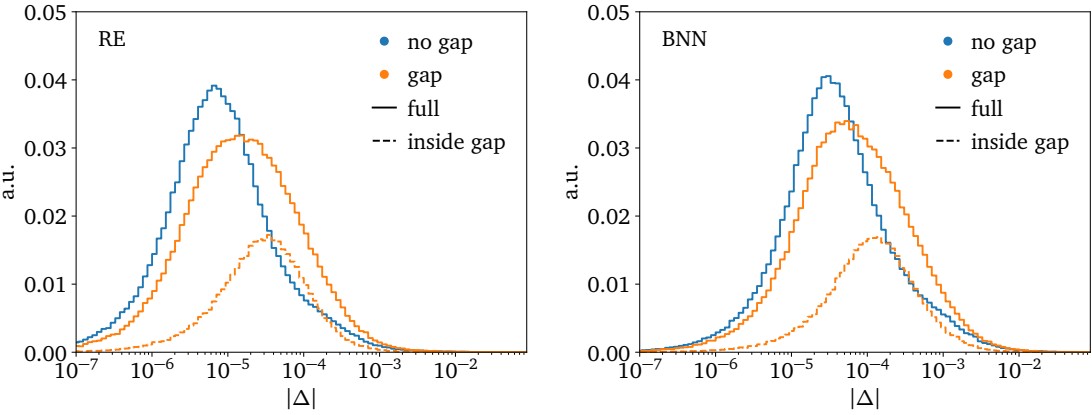

Figure 15: $|\Delta|$ distributions for the runs for the full training dataset (solid) and the events within the threshold gap (dashed). Left: repulsive ensemble results. Right: BNN results

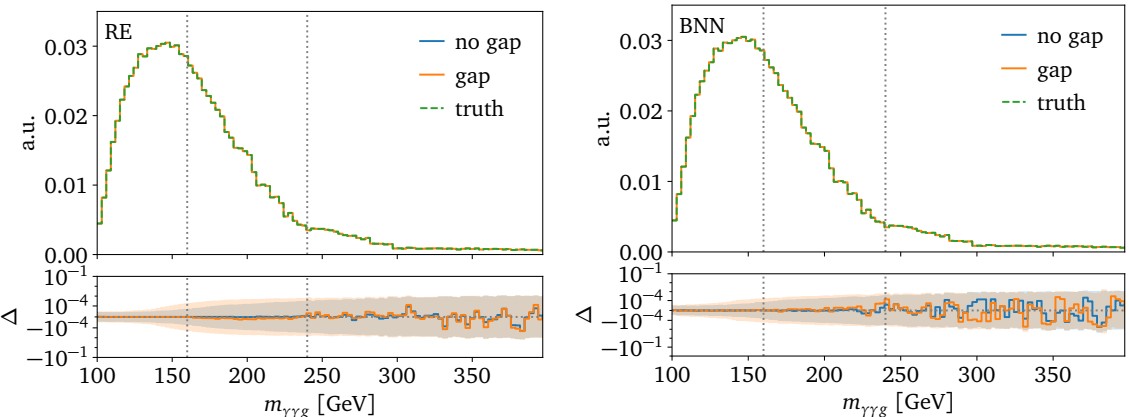

Figure 16: Upper panel: Invariant mass distribution of truth dataset (green), or reweighted using the surrogates trained on the dataset with (orange) or without (blue) gap. All three curves completely overlap. Lower panel: relative deviation from truth with uncertainty bands including the systematic and statistical uncertainties.

the effect is relatively modest for both approaches. Remarkably, the accuracy for the events in the threshold region (dashed orange curve) is not much worse than the overall accuracy for the whole test dataset (solid orange curve). This behavior highlights again the strong interpolation capability of neural networks, which can maintain reasonable accuracy even within regions not covered during training.

This is also visible in the $m_{\gamma\gamma g}$ distributions shown in the upper panels of Fig. 16, where we weight all events by $A_{\mathrm{NN}}(x)/A_{\mathrm{true}}(x)$ to emulate event generation with the trained surrogate. Again, we show the results for the repulsive ensemble and the BNN in the left and right plots, respectively. The shown truth curve overlaps with the prediction of both networks, whether trained on the full dataset or with the gap, thereby underpinning the previous result. We traced this behavior back to the amplitude being very flat in the considered $m_{\gamma\gamma g}$ region. We expect significantly worse predictions in cases with larger variations within the gap region.

Turning to the uncertainty estimate, the lower panel of Fig. 16 shows the relative deviation $\Delta$ together with the predicted total uncertainty indicated by shaded bands. While the average

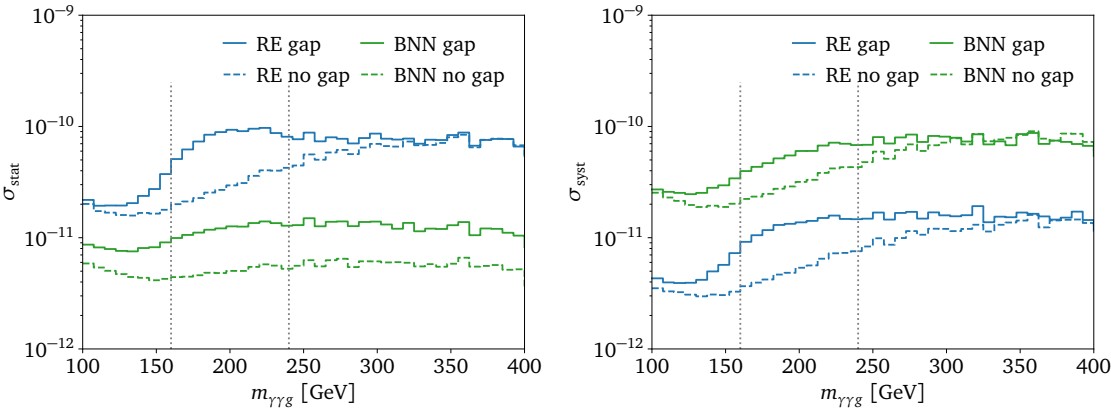

Figure 17: Left: Statistical uncertainty as a function of the invariant mass comparing repulsive ensemble and BNN results trained on the full or gap datasets. Right: Same as left, but the systematic uncertainty is shown. The gap region is indicated by dashed vertical lines.

deviation remains very small, i.e. $|\Delta| \lesssim 10^{-4}$, the uncertainty bands are considerably larger, pointing to an overestimation of the total uncertainty. As expected, the predicted uncertainty for the dataset with a gap is larger than for the dataset without a gap in the excluded region, although the difference is relatively modest.

We further investigate this behavior in the left panel of Fig. 17, showing the median statistical uncertainty as a function of $m_{\gamma\gamma g}$. Compared to the "no gap" result, we find that the repulsive ensemble correctly exhibits an increase of statistical uncertainty localized to the gap region, as indicated by the vertical lines. Outside the gap region, the uncertainty decreases again to the baseline level. For comparison, the BNN prediction shows a different pattern. Although the statistical uncertainty also increases inside the gap, it remains elevated across the entire invariant-mass range, indicating that the gap affects the BNN prediction more globally rather than locally. To test the robustness of these results, we also trained the surrogates with varying network sizes and different activation functions, finding similar results in all cases.

Naively, one might expect the statistical uncertainties predicted by the repulsive ensemble and the BNN to agree, since both approaches are based on the same underlying network architecture. In practice, however, two important aspects lead to differences. First, the training dynamics differ: even with the same model class, the two approaches converge to different minima and weight configurations, resulting in effectively different trained models. Second, the methods approximate the posterior in distinct ways, which directly affects how the statistical uncertainty is computed. These combined effects explain why the statistical uncertainties shown in Fig. 17 differ between the two approaches. We leave a more detailed understanding of these differences in a particle physics context for future work.

For the systematic uncertainty, shown in the right panel of Fig. 17, the behavior for the BNN and repulsive ensemble is similar to each other. Both architectures exhibit an increased systematic uncertainty in the gap region and outside the gap revert to the relative systematic uncertainty of the "no gap" case. However, while we can see that the repulsive ensemble estimates a larger statistical uncertainty than the BNN, the opposite is true for the systematic uncertainty. There, the BNN overall estimates a larger uncertainty compared to the repulsive ensemble. In both cases, the methods do not fully disentangle statistical from systematic effects, as seen from the rise of $\sigma_{\text{syst}}$ where ideally only $\sigma_{\text{stat}}$ should be affected.

We also tested evidential regression for the considered gap scenario. While it shows equally good interpolation capabilities in the gap region, we find the estimated statistical and systematic uncertainty to be a flat function of $m_{\gamma\gamma g}$.

# 6 Conclusions

Surrogate amplitudes are an important ingredient for speeding up high-precision Monte Carlo event generation. The key requirements are speed, precision, and control. In this paper, we have worked towards these goals by investigating three different approaches — repulsive ensembles, evidential regression, and Bayesian neural networks — and testing their behavior in scenarios with locally noisy or missing data.

Repulsive ensembles are a collection of networks including a repulsive interaction. The spread of ensemble members provides an approximation of the posterior predictive distribution, thereby serving as a measure of statistical uncertainty. We first studied how the strength of the repulsive interaction affects amplitude prediction and uncertainty estimation, finding its effect to be negligible for sufficiently large datasets. Moreover, we studied whether ensembling improves accuracy. While it reduces noise in the network predictions, it does not alleviate systematic biases. Building on this, we revisited the miscalibration of systematic uncertainties identified in earlier work. We traced this to a mismatch: while the ensemble mean prediction is more accurate than the individual members, the corresponding mean of the uncertainty estimates does not improve in the same way, leading to miscalibration. To address this, we proposed a method to learn a systematic uncertainty directly for the ensemble mean prediction. This approach yielded well-calibrated uncertainties for small ensemble sizes. For larger ensembles, however, it indicated residual biases. We traced these back to non-Gaussian effects that are not captured by the Gaussian ansatz used in the likelihood. Such issues can be mitigated by employing more expressive networks, improved training strategies, or a more general likelihood formulation.

In addition to repulsive ensembles, we investigated evidential regression as an alternative approach that encodes all uncertainties directly in the network outputs, without requiring an ensemble. This method is computationally more efficient, and for the unsmeared amplitude dataset, it gave results consistent with the repulsive ensemble approach. Moreover, when comparing two variants of evidential regression, we observed that constraining two of the network outputs, i.e. $\alpha = 2\nu$, outperforms the version with an additional regularization loss.

Afterwards, we investigated whether the trained networks can capture localized noise or gaps in the dataset — mimicking numerical instabilities in amplitude evaluations, such as near particle thresholds — and appropriately quantify this through their predicted uncertainties. Focusing first on smearing in a small, box-shaped region of the invariant mass distribution, we found that both methods can effectively identify and describe this region. While the repulsive ensemble followed the expected behavior of the systematic uncertainty more closely, the evidential regression and Bayesian neural networks provided a better calibration of the uncertainty. As a next step, we investigated a smearing effect that becomes increasingly pronounced near a particle mass threshold. Here again, all approaches followed the expected behavior very well, with the repulsive ensemble and Bayesian neural networks slightly outperforming the evidential regression approach. Finally, we considered a data gap in the invariant mass distribution, i.e. a localized region in which no training data is provided. Despite the absence of data, the networks produced good predictions in the gap because the amplitude varies only slowly in that region. As expected, repulsive ensembles and Bayesian neural networks predicted an increased uncertainty in the gap region.

Overall, we have extended our toolkit and deepened our understanding of amplitude surrogates. Repulsive ensembles capture uncertainty more reliably but at a higher computational cost, while evidential regression is more efficient and can yield well-calibrated uncertainties in specific scenarios. These insights guide the future development of robust surrogate models for next-generation Monte Carlo event generators.

# Acknowledgements

We thank Thomas Gehrmann for the fruitful discussion on numerical noise in the calculation of scattering amplitudes. NE is funded by the Heidelberg IMPRS *Precision Tests of Fundamental Symmetries*. This research is supported through the KISS consortium (05D2022) funded by the German Federal Ministry of Education and Research BMBF in the ErUM-Data action plan,by the Deutsche Forschungsgemeinschaft (DFG, German Research Foundation) under grant 396021762 – TRR 257: *Particle Physics Phenomenology after the Higgs Discovery*, and through Germany's Excellence Strategy EXC 2181/1 – 390900948 (the *Heidelberg STRUCTURES Excellence Cluster*). Finally, we would like to thank the Baden-Württemberg Stiftung for financing through the program *Internationale Spitzenforschung*, project *Uncertainties – Teaching AI its Limits* (BWST_ISF2020-010).

# A  Non-linear error propagation

When training the network, we actually fit the preprocessed amplitudes, obtained by applying a logarithm and subsequent standardization,

$$\ell_{\text{train}}(x) = \frac{\log A_{\text{train}}(x) - \mu_{\text{train}}}{s_{\text{train}}}$$

$$\text{with} \qquad \mu_{\text{train}} = \big\langle \log A_{\text{train}}(x) \big\rangle \qquad s_{\text{train}} = \sqrt{\text{Var}(\log A_{\text{train}}(x))} \,. \tag{57}$$

The network then predicts for each phase-space point $x$

$$\text{NN}(x, \theta) = \begin{pmatrix} \bar{\ell}(x, \theta) \\ \log \sigma_\ell^2(x, \theta) \end{pmatrix}, \tag{58}$$

where $\bar{\ell}(x, \theta)$ denotes the predicted mean of the standardized log-amplitude and $\sigma_\ell^2(x, \theta)$ its variance. Averaging over the weight posterior approximation $q(\theta)$ as in Eq.(2), the predictive mean and variance in $\ell$-space are given by

$$\ell_{\text{NN}}(x) = \int \text{d}\theta \, q(\theta) \bar{\ell}(x, \theta)$$

$$\sigma_{\ell,\text{tot}}^2(x) = \int \text{d}\theta \, q(\theta) \left[ \sigma_\ell^2(x, \theta) + \big(\bar{\ell}(x, \theta) - \ell_{\text{NN}}(x)\big)^2 \right], \tag{59}$$

with the usual decomposition

$$\sigma_{\ell,\text{syst}}^2(x) = \int \text{d}\theta \, q(\theta) \, \sigma_\ell^2(x, \theta), \qquad \sigma_{\ell,\text{stat}}^2(x) = \int \text{d}\theta \, q(\theta) \big(\bar{\ell}(x, \theta) - \ell_{\text{NN}}(x)\big)^2. \tag{60}$$

Transforming back to amplitude space, the inverse for a single network pass is

$$\overline{A}(x, \theta) = \exp\big(s_{\text{train}} \bar{\ell}(x, \theta) + \mu_{\text{train}}\big) \,. \tag{61}$$

However, we are interested is the predictive mean $A_{\text{NN}}(x)$ after averaging over $q(\theta)$. Because the inverse mapping is a non-linear function, we must propagate the log-space uncertainty explicitly. Assuming that $\bar{\ell}(x, \theta)$ is Gaussian distributed with mean $\ell_{\text{NN}}(x)$ and variance $\sigma_{\ell,\text{tot}}^2(x)$, following Ref. [62] we obtain

$$A_{\text{NN}}(x) = \int \text{d}\bar{\ell} \, \overline{A}(x, \theta) \, \mathcal{N}\big(\bar{\ell} \,\big|\, \ell_{\text{NN}}(x), \sigma_{\ell,\text{tot}}(x)\big)$$

$$= \int \text{d}\bar{\ell} \, \exp\big(s_{\text{train}} \bar{\ell} + \mu_{\text{train}}\big) \, \mathcal{N}\big(\bar{\ell} \,\big|\, \ell_{\text{NN}}(x), \sigma_{\ell,\text{tot}}(x)\big)$$

$$= \exp\left( s_{\text{train}} \ell_{\text{NN}}(x) + \mu_{\text{train}} + \frac{s_{\text{train}}^2 \sigma_{\ell,\text{tot}}^2(x)}{2} \right)$$

$$\approx \exp(s_{\text{train}} \ell_{\text{NN}}(x) + \mu_{\text{train}}) \,, \tag{62}$$

where the last line of approximation holds for $s_{\text{train}}^2 \sigma_{\ell,\text{tot}}^2 \ll s_{\text{train}} \ell_{\text{NN}}(x)$. In the same way, we can then calculate the total predictive uncertainty of the amplitude $A$ as given by

$$\sigma_{A,\text{tot}}^2(x) = \int \text{d}\bar{\ell} \, \big(\overline{A}(x, \theta) - A_{\text{NN}}(x)\big)^2 \, \mathcal{N}\big(\bar{\ell} \,\big|\, \ell_{\text{NN}}(x), \sigma_{\ell,\text{tot}}(x)\big)$$

$$= A_{\text{NN}}^2(x) \left[ \exp\big(s_{\text{train}}^2 \sigma_{\ell,\text{tot}}^2(x)\big) - 1 \right]$$

$$\approx s_{\text{train}}^2 \, A_{\text{NN}}^2(x) \, \sigma_{\ell,\text{tot}}^2(x) \,. \tag{63}$$

We note that the approximate formula in the last line recovers the standard linearized error propagation formula.

Beyond the linearized regime, the decomposition into systematic and statistical parts in amplitude space is no longer strictly additive. We can first define

$$\nu_{\text{tot}} = s^2_{\text{train}} \sigma^2_{\ell,\text{tot}} \qquad \nu_{\text{syst}} = s^2_{\text{train}} \sigma^2_{\ell,\text{syst}} \qquad \nu_{\text{stat}} = s^2_{\text{train}} \sigma^2_{\ell,\text{stat}} . \tag{64}$$

Then expanding to second order in $\nu_{\text{tot}}$ yields

$$\frac{\sigma^2_{A,\text{tot}}(x)}{A^2_{\text{NN}}(x)} \approx \nu_{\text{syst}}(x) + \nu_{\text{stat}}(x) + \tfrac{1}{2}\Big[\nu^2_{\text{syst}}(x) + \nu^2_{\text{stat}}(x)\Big] + \underbrace{\nu_{\text{syst}}(x)\,\nu_{\text{stat}}(x)}_{\text{interaction}} . \tag{65}$$

In the regime relevant for this work, however, the log-space variances are extremely small, with typical values $\nu_{\text{tot}} \sim \mathcal{O}(10^{-10})$. Consequently, all quadratic and interaction terms are suppressed by many orders of magnitude relative to the linear contributions.

We have explicitly verified this by evaluating the full expression above and comparing it to the linear approximation used in the main text, finding relative differences well below numerical precision. We therefore conclude that, for all results presented in this paper, the exponential back-transformation operates entirely in the linear regime and does not affect the decomposition or interpretation of statistical and systematic uncertainties.

## B Hyperparameters

Throughout this work, we use the same hyperparameter settings compiled in Tab. 2. The settings are taken over from Ref. [78], in which the effect of different choices is discussed in detail.

| Parameter | Value |
|---|---|
| Activation function | GELU |
| Number of hidden layers | 6 |
| Hidden nodes | 128 |
| Batch size | 1024 |
| Scheduler | One cycle |
| Max learning rate | $10^{-3}$ |
| Number of epochs | 1000 |

Table 2: Network and training parameters.

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
