# Peer review of "Amplitude Uncertainties Everywhere All at Once"

_SciPost Physics_

## Round 1 · Referee Report · Anonymous (Referee 1) · 2025-11-21

Strengths

  1. The paper presents a thorough investigation of uncertainty estimation for amplitude surrogates.

  2. The authors considers different scenarios that cam impede the training and accuracy of the surrogate models, including localised training data inaccuracies or missing data

Weaknesses

  1. Section 3 reports a bias without investigating (or reporting on the investigation) of its possible source

Report

The manuscript presented is an investigation worthy of publication, provided the requested changes are addressed. I enjoyed reviewing it.

Requested changes

Can the authors please

  • specify how many training were performed for the bands in Fig 2? The manuscript only mentions "multiple times".

  • use a logarithmic scale for the upper panels of Figure 2

  • investigate the source of the bias reported in section 3:

Section 3.2 reports and investigates a bias in the ensemble method. In my opinion this bias is likely due to the fact that the authors fit the logarithm of the amplitude and not the amplitude itself. The effect of this transformation is investigated in their appendix A, from there is is clear that fitting the logarithm of the amplitude and transforming back will yield a positive bias in the amplitude. Can the author check whether this is the source of the bias or clearly exclude this possibility by reporting the values in what they called "l-space" and possibly reporting the value of the bias induced by the transformation alongside the one they measure?

The transformation also makes the discussion in 3.3 more difficult: sigma_stat is not applying to the amplitude but to its logarithm. I would question the validity of the derivation. The following section could be the solution to an inexistant problem.

Can the authors also clarify whether the ensemble average for the amplitude is obtained by exponentiating the average of the logarithm prediction or by averaging the exponentiated logarithms?

  • check whether the scale of the y axis of figure 4 right? It would indicate that the mean relative error for amplitudes > 10^5 is larger than 100%?

  • compare the "bias floor" they found in figure 5 with that expected from the logarithmic transformation?

  • clarify the meaning of "channel" in the caption of figures 5 and 6.

  • make the dashed blue line in figure 5 more visible (or state behind which other line it is hiding)

  • elaborate on the discussion on why \sigma_syst should converge to |A_NN-A_train|, the reader is pointed to an explaation in section 2.1, but I could not find one. If the authors mean to refer to the explanation in Appendix D of their reference 65, perhaps they can remove this level of indirection. The description in that appendix refers to this effect as "ideally ..." so it would be useful for the authors to justify why the formuation of the globally learned systematic uncertainty allows for this ideal case while other strategies do not.

Recommendation

Ask for minor revision

---

## Round 1 · Referee Report · Anonymous (Referee 2) · 2025-12-16

Strengths

  1. the paper provides a good comparison of three distinct uncertainty quantification approaches (repulsive ensembles, evidential regression, compared to BNNs), including their computational trade-off and performances.

  2. the problem of reliable uncertainty estimation for amplitude surrogates is crucial for next-generation Monte Carlo generators, and the paper addresses important challenges that arise in these settings.

  3. the identification and analysis of the miscalibration issue in repulsive ensembles discussed in Sect.3.3 is valuable, has clear mathematical derivation showing why averaging individual uncertainties fails for model-error dominated regimes.

  4. the proposed method of learning a global systematic uncertainty for the ensemble mean is interesting and addresses a real limitation of standard ensemble approaches. Also, the investigation of threshold smearing and data gaps in Sect. 5 demonstrates practical usage and reveals important differences between methods.

  5. the paper is well-written and well-structured, moving from theoretical foundations to practical tests in more than one scenarios.

Weaknesses

  1. the analysis relies on a single process and the generalisability of the conclusions remains unclear, as different processes with varying amplitude structures could behave differently.

  2. some choices in the parameters used in the various approaches are a bit arbitrary, although they provide a good starting point for further investigation.

  3. the more realistic scenarios presented by the authors in sect. 5 are more challenging but perhaps still too far from realistic scenarios, however once again they provide a good starting point for further investigation.

Report

This paper investigates uncertainty quantification methods for neural network surrogates of high-energy physics scattering amplitudes, taking gg > gam gam g as a case study and focussing on repulsive ensembles, evidential regression, and Bayesian neural networks. The authors examine calibration issues, propose solutions for systematic uncertainty estimation, and test these methods in various settings. The work is technically solid and addresses an important practical problem for future LHC event generation. The paper is well written and clear. I recommend it to be published subject to the minor corrections I suggest in my report.

Requested changes

(i) in sect. 2.1, after Eq. (5) it would be good to anticipate the discussion on the validity of the simple Gaussian assumption and how this is going to be tested throughout the paper. Also there is some confusion about A_train and A_true. It seems to me that the authors always use the MC simulation to train the surrogate NN model, can the authors clearly highlight the differences between A_train and A_true?

(ii) in sect. 2.2, when the GMM is introduced, it would be good to anticipate what is the typical number of modes that is going to be explored (K in the notation adopted in the paper)

(iii) in sect. 2.3, some settings are taken for granted. For example, how are the 1000 epochs justified, are 1000 epochs enough? Also, what is the typical number of batches B? Moreover, and more importantly, it would be good to show here a plot of the typical size of the amplitude (for example a histogram of the magnitude of A_train across the phase space points x that are explored in this study) as this would give an idea of how significant is the bias that the authors find later (for large values of the amplitude)

(iv) in sect. 3.1 I find the N_train vector a bit confusing. The number of training point is not particularly significant for an external reader, it would be better to have the percentage of the full dataset in the training set.

(v) in the plots on the left hand side of Fig. 3 it is not clear to me why the histogram of the mean is away from the histograms of the individual ensembles.

(vi) around Eq. (36) it is not clear how \phi is to be determined. Also, it would be interesting to understand whether the bias floor shown in Fig. 5 is still dominated by the inability of the NNs to represent large amplitudes.

(vii) in Fig 6 the non-Gaussian residuals observed for large ensembles, N_ens>100 suggest the Gaussian likelihood assumption breaks down, but this is not thoroughly addressed, the authors could perhaps add some comments when discussing Fig 6.

(viii) after Eq. (48) how is the choice of \lambda =0.01 justified? Analogously, after Eq. (50) how is r =1 chosen?

(ix) In sect.5, why is the threshold set at 200 GeV? I find the discussion of the results in the various scenarios very interesting. I think it would help the reader to add a table summarising the performance of the three explored methodologies (repulsive ensembles, evidential regression and BNN) in the three scenarios that are explored.

Recommendation

Ask for minor revision

---

## Editorial Decision

in_refereeing